# The CNES Solutions for Improving the Positioning Accuracy with Post-Processed Phase Biases, a Snapshot Mode, and High-Frequency Doppler Measurements Embedded in Recent Advances of the PPP-WIZARD Demonstrator

Clément Gazzino [1,*], Alexis Blot [1], Elodie Bernadotte [1], Théo Jayle [2], Marion Laymand [2], Nicolas Lelarge [2], Aude Lacabanne [2] and Denis Laurichesse [1]

1   Centre National d'Études Spatiales—CNES, 31400 Toulouse, France; alexis.blot@cnes.fr (A.B.); elodie.bernadotte@cnes.fr (E.B.); denis.laurichesse@cnes.fr (D.L.)
2   CS Group, 31400 Toulouse, France; theo.jayle@csgroup.eu (T.J.); marion.laymand@csgroup.eu (M.L.); aude.lacabanne@csgroup.eu (A.L.)
*   Correspondence: clement.gazzino@cnes.fr

**Abstract:** For many years, the navigation team at the French Space Agency (CNES) has been developing its Precise Point Positioning project. The goal was initially to promote a technique called undifferenced ambiguity resolution. One of the main characteristics of this technique is the capability for a user receiver to perform centimeter-level accuracy in real time. To do so, a demonstrator has been built. Its architecture is composed of three main elements: a correction processing software called the server part, a means to transmit the corrections using standardized messages, and a user software capable of handling the corrections to compute an accurate positioning at the user level. In this paper, we present the recent advances in the CNES precise point positioning demonstrator. They are composed of some evolution of the network of stations and server software, the implementation of the new state space representation standard, a new method for instantaneous ambiguity resolution using uncombined four-frequency signals, its implementation in real-time at the server and the user level, and the use of high-rate Doppler measurements to improve the accuracy of the solution in harsh urban environments. On top of that, the computation of high-accuracy post-processed phase biases with the majority of current GNSS signals supported, compatible with the uncombined method and a new online positioning service to demonstrate the capacity of the user software, is demonstrated.

**Keywords:** precise point positioning; real-time; ambiguity resolution; multi frequency GNSS; online service; post-processed biases; Doppler shift measurements

## 1. Introduction

The measure of the carrier phase of the signals transmitted by the global satellite navigation systems (GNSS) enables, together with the pseudo range and the Doppler shift measurements, a precise positioning to a centimeter range accuracy (see [1] (Chapter 8) and the references therein) assuming that all error sources can be mitigated and the carrier phase ambiguity is determined. This opened the door for the Real-Time Kinematic (RTK) and the Precise Point Positioning (PPP) techniques for high-precision positioning [2] (Chapter 8). The PPP technique is powerful because it does not require a costly infrastructure of stations such as the RTK one, and hence has been widely used from several decades up to recent applications, either public or commercial [3–8]. However, a major drawback of PPP is the convergence time. Dual-frequency PPP convergence is long, in the order of tens of minutes, which makes it impracticable for many applications. Until now, some improvements have been made using a third frequency (L1/L2/L5 for GPS, E1/E5a/ E5b for Galileo).

It is now widely accepted that PPP techniques can achieve centimeter-level accuracy globally in real time, in particular when they are combined with phase integer ambigu-

ity resolution (AR). To solve for the integer ambiguities, the reference [9] proposed an integer clocks approach and the reference [10] a decoupled clock model. Thanks to these approaches, computing differences between satellites are avoidable for the users, contrary to the original technique from Ge [11]. A prerequisite for integer AR on the user side is the need for a ground stations network for computing precise satellite orbits as well as integer recovery clocks or uncalibrated phase delays in the case of an ionosphere-free dual frequency combination. The adoption of a third and a fourth frequency by recent satellites of the GNSS constellations also requires the knowledge of the interfrequency code biases (IFCB). Recent studies concluded that the goal of centimeter-level accuracy positioning can be achieved after the convergence phase even in real time [12]. In the past few years, the real-time network of stations has evolved constantly. The majority of the GNSS observables collected at each ground station are available through the Intenational GNSS Service (IGS) and the Real-Time Service (RTS) [13]. In particular, the French Centre National d'Etudes Spatiales (CNES) contributes significantly with its own network [14]. Other networks are also available, such as AUSCORS, EUREF, and UNAVCO. It is important to note that the modernization of the stations quickly leads to full GNSS capability, enabling real-time dissemination of the GNSS measurements. Using these real-time measurements, CNES has implemented the method for computing code and phase biases described in [9] and built a real-time demonstrator allowing the dissemination of these code and phase biases as well as orbit and clock corrections [15] called PPP-WIZARD (PPP With Zero-difference Ambiguity Resolution Demonstrator). The biases are broadcast following the Radio Technical Commission for Maritime Services (RTCM) definition, and, in 2020, the IGS decided to define its own State Space Representation (SSR) messages standard, mainly to fill some gaps in the RTCM one, such as the lack of phase-bias messages. This standard is implemented in the CNES demonstrator both on the network side and the user side. Since 2020, the corrections are broadcast in both RTCM and IGS standards. We characterize this bias in terms of noise and time stability and show that it can be casted into the SSR representation proposed by the RTCM or the IGS for phase-bias messages.

In July 2018, a collaboration between NRCan (Natural Resources of Canada) and CNES concluded that instantaneous (epoch-by-epoch) convergence is possible thanks to the use of the new E6 Galileo signal [16]. An accuracy of 15 cm is achievable using Galileo alone, and centimeter-level accuracy can be obtained by combining Galileo and GPS measurements. In this paper, the real-time implementation of this new method called OEUFS (Optimal Estimation using Uncombined Four-frequency Signals) in the CNES PPP demonstrator is presented. We investigate deeper the benefits of using this method through the PPP-Snapshot concept with the instantaneous convergence allowed by the four frequencies case, as apposed to what has been shown in [16]. Improvements are also achievable by adding high-rate Doppler measurements. The user-level implementation is then described. Since OEUFS involves optimal partial ambiguity resolution, the BIE (Best Integer Equivariant) method has been integrated. Some actual real-time positioning results are presented in good conditions (open sky) and in difficult environments, such as dense urban areas with narrow streets or medium urban areas with foliage.

Despite the fact that the number of Galileo satellites and E6-compatible stations at the IGS RTS is growing, the real-time phase-biases production suffers from some limitations such as the lack of compatible stations or the fact that some BDS-3 real-time observables are not available. To overcome these drawbacks, we describe how a post-processed phase-biases solution can be designed and computed. The idea is to take an IGS solution (orbits, clocks, code biases, yaw ...) and to compute a set of phase biases consistent with this solution using the majority of the stations available at the IGS (around 250). We can then obtain an optimal and accurate set of phase biases and better demonstrate the instantaneous ambiguity resolution. In this paper, we show how to compute such a solution. The main idea is to perform some carefully chosen phase-residuals combinations to solve for the ambiguities in these combinations, then to identify the corresponding biases, and finally to recover the individual phase biases by inverting a system of equations. The code and

phase biases for all GNSS signals are computed except for the GLONASS phase biases. Indeed, the Frequency Division Multiple Access (FDMA) technique to distinguish among the signals broadcast by the Russian constellation prevents us from solving for the integer ambiguities. This process is done on a daily basis, and the products are freely available for test purposes.

Finally, the combination of post-processed phase biases and the user software allows very accurate PPP solutions to the centimeter level. This article proposes some comparisons with other existing services, either academic or commercial. To demonstrate such a capacity, we propose an online positioning service. This service works similar to the other services available on the internet [17–19]: the user uploads a rinex file along with a small set of configuration parameters, such as static or kinematic positioning, and the service returns the PPP trajectory, in an NMEA file in the kinematic case or the cartesian coordinates in the static case. The majority of the features of the demonstrator, such as the multi-constellation and multi-frequency ambiguity resolution, are supported. The targeted accuracy for this service is the centimeter.

## 2. The CNES PPP-WIZARD Demonstrator

The PPP-WIZARD (PPP with Zero-difference Ambiguity Resolution Demonstrator) was first designed to promote the undifferenced phase ambiguity resolution technique developed at CNES [9,20]. This proof-of-concept for the aforementioned original ambiguity resolution technique aims at achieving a Precise Point Positioning (PPP) accuracy of 1 cm in real time. It relies on the means available at the IGS [13,21] and in particular its Real-Time Service (RTS) [22], such as the access to a network of GNSS stations or the standards for the dissemination of the corrections. The demonstrator is also a good laboratory to test new ideas related to PPP with ambiguity resolution.

The overall PPP approach is depicted in Figure 1 and involves the following steps:

1. on the network side, also named server side, raw data are collected thanks to a global network of stations, then the main processing software Orbit Determination and Time Synchronization (ODTS) computes all the necessary corrections (orbits, clocks, biases, ionosphere) that are compatible with the ambiguity resolution on the user side;
2. the corrections are disseminated over the network in an open standard, RTCM, or SSR;
3. on the user side, the PPP-User software estimates the position of the receiver by means of a stochastic filter, leading to centimeter-level PPP by fixing the integer ambiguities of the phase measurements.

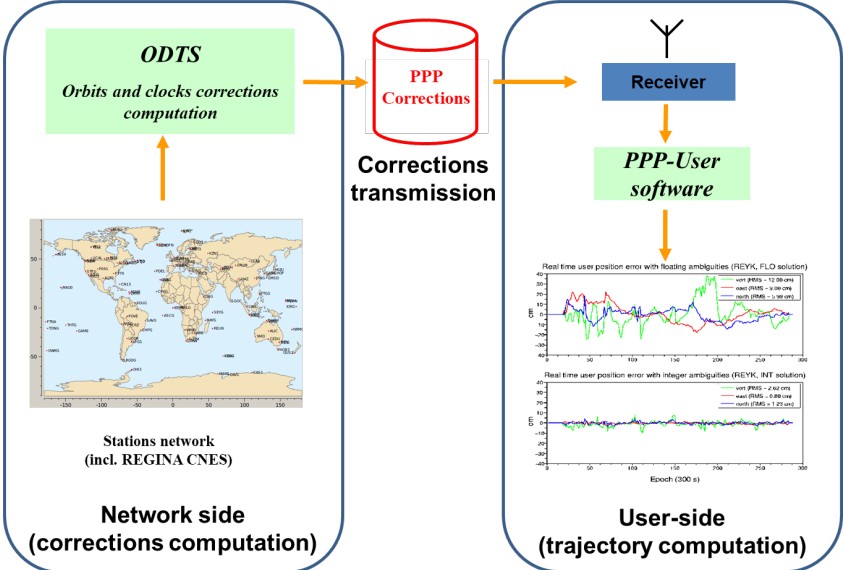

**Figure 1.** The PPP-WIZARD overall architecture.

### 2.1. The Network Side of the Demonstator

The network of stations, from which the raw data are collected, is constantly evolving. We present here the network used in June 2023. It consists of 154 stations evenly distributed around the globe (see Figure 2), from the following networks: IGS (118 stations), AUSCORS (11 stations), UNAVCO (23 stations), EUREF (2 stations). Some of the IGS stations are part of the CNES REGINA network [14]. The majority of this network is now compatible with the modernized GNSS (multi-constellations, multi-frequencies).

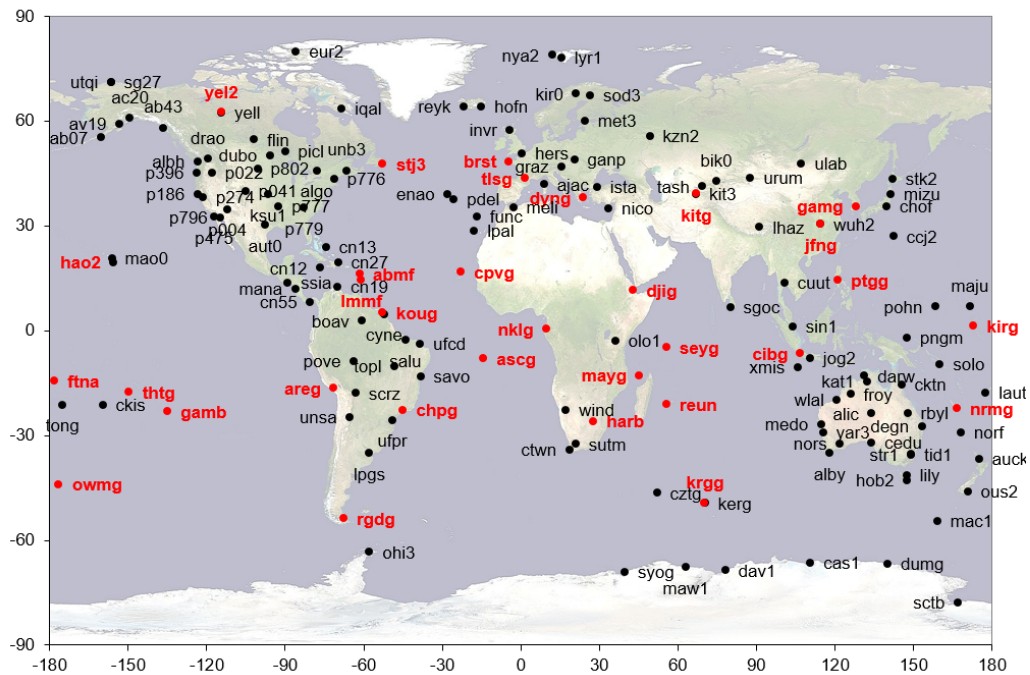

**Figure 2.** The PPP-WIZARD network of stations. Red stations are part of the CNES REGINA network [14] and black stations are part of the other networks (IGS, AUSCORS, UNAVCO, and EUREF).

After recovering data from the ground stations, the Orbit Determination and Time Synchronization (ODTS) software computes the corrections. This software is described in reference [15], and the method used for the computations is described in references [9,23–26] and the references therein. It is composed of several Kalman filters. A subset of filters is dedicated to various widelane bias computations, whereas a main filter performs the clock computation.

The multi-frequency code and phase measurements from a GNSS satellite are modeled as follows: for a receiver obtaining the signal from a satellite $s$ at the frequency $f_j$ (and wavelength $\lambda_j$), the code and phase measurements on this frequency are given by:

$$
\begin{aligned}
C_i &= \rho_r^s + h_r^s + \gamma_i e + m(E^s)T_Z + b_{r,C_i}^s, \\
\lambda_i L_i &= \rho_r^s + h_r^s - \gamma_i e + m(E^s)T_Z + \lambda_i N_i^s + \lambda_i W + b_{r,L_i}^s,
\end{aligned}
\tag{1}
$$

where $C_i$ is the code measurement expressed in the unit of distance, $L_i$ is the carrier phase measurement expressed in the unit of cycles at the frequency $f_i = c/\lambda_i$, $c$ is the speed of light in vacuum, $h_r^s = h_r - h^s$ is the clock offset between the satellite and the receiver, $e$ is the slant ionospheric elongation at the reference frequency $f_1 = 1575.42$ MHz (corresponding to the L1 / E1 signals), $\gamma_i = f_1^2/f_i^2$, $T_z$ is the zenithal wet tropospheric delay, $m(E^s)$ is the mapping function depending on the satellite elevation $E^s$ of the satellite $s$, $W$ is the phase wind-up effect, and $N_i^s$ is the carrier phase ambiguity of the signal coming from the satellite $s$ at the frequency $f_i$. The difference between the code and phase biases at the satellite



and receiver levels on the frequency $f_i$ are denoted, respectively, as $b_{r,C_i}^s = b_{r,C_i} - b_{C_i}^s$ and $b_{r,L_i}^s = b_{r,L_i} - b_{L_i}^s$. $\rho_r^s$ is the geometric distance between the satellite $s$ and the receiver $r$ including the phase center offset correction and is computed as the scalar product between $\mathbf{u}_r^s$, the unit-vector of the satellite–receiver line-of-sight, and $\Delta \mathbf{p}_r^s$, the ranging vector from the receiver to the satellite:

$$\rho_r^s = \mathbf{u}_r^s \cdot \Delta \mathbf{p}_r^s + PCO_j, \tag{2}$$

with $PCO_j$ being the phase center offset for the frequency $f_j$.

The ionosphere-free (IF) combinations between the code and phase measurements of signals at the frequencies $f_i$ and $f_j$ are given by:

$$
\begin{aligned}
C_{\mathrm{IF},ij} &= \frac{\gamma_j C_i - \gamma_i C_j}{\gamma_j - \gamma_i} = \rho_r^s + h_r^s + m(E)T_Z + \frac{\gamma_j b_{r,C_i}^s - \gamma_i b_{r,C_j}^s}{\gamma_j - \gamma_i}, \\
\lambda_{\mathrm{IF},ij} L_{\mathrm{IF},ij} &= \frac{\gamma_j \lambda_i L_i - \gamma_i \lambda_j L_j}{\gamma_j - \gamma_i} = \rho_r^s + h_r^s + m(E)T_Z + \frac{\gamma_j b_{r,L_i}^s - \gamma_i b_{r,L_j}^s}{\gamma_j - \gamma_i} + \frac{\gamma_j \lambda_i N_i^s - \gamma_i \lambda_j N_j^s}{\gamma_j - \gamma_i} + \frac{\gamma_j \lambda_i - \gamma_i \lambda_j}{\gamma_j - \gamma_i} W,
\end{aligned}
\tag{3}
$$

with $\lambda_{\mathrm{IF},ij} = \dfrac{\gamma_j \lambda_i - \gamma_i \lambda_j}{\gamma_j - \gamma_i}$ being the IF wavelength.

As defined in [27], the ionosphere-free clock offsets for the code and phase measurements can be formulate as:

$$
\begin{aligned}
h_{r,\mathrm{IF},C_{ij}}^s &= h_r^s + \frac{\gamma_j b_{r,C_i}^s - \gamma_i b_{r,C_j}^s}{\gamma_j - \gamma_i}, \\
h_{r,\mathrm{IF},L_{ij}}^s &= h_r^s + \frac{\gamma_j b_{r,L_i}^s - \gamma_i b_{r,L_j}^s}{\gamma_j - \gamma_i},
\end{aligned}
\tag{4}
$$

as well as their satellite and receiver counterparts $h_{r,\mathrm{IF},C_{ij}}^s = h_{r,\mathrm{IF},C_{ij}} - h_{\mathrm{IF},C_{ij}}^s$ and $h_{r,\mathrm{IF},L_{ij}}^s = h_{r,\mathrm{IF},L_{ij}} - h_{\mathrm{IF},L_{ij}}^s$. Thanks to these IF clocks, the satellite–receiver clock offset and signal biases are transformed to:

$$
\begin{aligned}
h_r^s + b_{r,C_i}^s &= h_{r,\mathrm{IF},C_{ij}}^s + \gamma_i \tau_{r,C_{ij}}^s, & \tau_{r,C_{ij}}^s &= \frac{b_{r,C_j}^s - b_{r,C_i}^s}{\gamma_j - \gamma_i}, \\
h_r^s + b_{r,L_i}^s &= h_{r,\mathrm{IF},L_{ij}}^s - \gamma_i \tau_{r,L_{ij}}^s, & \tau_{r,L_{ij}}^s &= \frac{b_{r,L_j}^s - b_{r,L_i}^s}{\gamma_j - \gamma_i},
\end{aligned}
\tag{5}
$$

and the code and phase measurement models can be expressed with respect to the code and phase IF clocks, respectively:

$$
\begin{aligned}
C_i &= \rho_r^s + \gamma_i e + m(E^s)T_Z + h_{r,\mathrm{IF},C_{ij}}^s + \gamma_i \tau_{r,C_{ij}}^s, \\
C_j &= \rho_r^s + \gamma_j e + m(E^s)T_Z + h_{r,\mathrm{IF},C_{ij}}^s + \gamma_j \tau_{r,C_{ij}}^s, \\
\lambda_i L_i &= \rho_r^s - \gamma_i e + m(E^s)T_Z + \lambda_i N_i^s + \lambda_i W + h_{r,\mathrm{IF},L_{ij}}^s - \gamma_i \tau_{r,L_{ij}}^s, \\
\lambda_j L_j &= \rho_r^s - \gamma_j e + m(E^s)T_Z + \lambda_j N_j^s + \lambda_j W + h_{r,\mathrm{IF},L_{ij}}^s - \gamma_j \tau_{r,L_{ij}}^s.
\end{aligned}
\tag{6}
$$

This model follows the IGS standards. $\tau_{r,C_{ij}}^s$ plays the role of a Time Group Delay (TGD) and $\tau_{r,L_{ij}}^s$ is its counterpart for the carrier-phase measurement.

It is well known that the ambiguity $N_{\mathrm{WL},ij} = N_i - N_j$ of the widelane combination of the carrier phases $L_i$ and $L_j$ has a low noise level compared to the individual $N_i$ and $N_j$ ambiguities thanks to the larger wavelength of the widelane. In order to let this widelane ambiguity appear in the models, we compute the Melbourne–Wübbena [28,29], combination between the signals at the frequencies $f_i$ and $f_j$. This combination consists of the sum

of the widelane carrier phase combination and the narrowlane code combination and reads:

$$\text{MW}(L_i, L_j, C_i, C_j) = L_i - L_j + \frac{\lambda_i - \lambda_j}{\lambda_i + \lambda_j}\left(\frac{C_i}{\lambda_i} + \frac{C_j}{\lambda_j}\right). \tag{7}$$

Replacing Equation (6) in Equation (7) leads to:

$$\text{MW}(L_i, L_j, C_i, C_j) = N_{\text{WL},ij} + \mu^s_{r,ij}, \tag{8}$$

where $\mu^s_{r,ij}$ can be interpreted as a widelane fractional bias expressed in cycles by:

$$\mu^s_{r,ij} = \left(\frac{1}{\lambda_i} - \frac{1}{\lambda_j}\right)\left(h^s_{r,\text{IF},L_{ij}} - h^s_{r,\text{IF},C_{ij}}\right) + \left(\frac{\gamma_i}{\lambda_i} - \frac{\gamma_j}{\lambda_j}\right)\left(\tau^s_{r,L_{ij}} + \frac{\lambda_i - \lambda_j}{\lambda_i + \lambda_j}\tau^s_{r,C_{ij}}\right), \tag{9}$$

which can be decomposed into its satellite and its receiver parts: $\mu^s_{r,ij} = \mu_{r,ij} - \mu^s_{ij}$.

It has been shown in [25] that for receivers in good environmental conditions, the receiver biases are generally stable over a satellite pass. It has to be noted that the receivers' and satellites' widelane phase biases can be defined modulo an integer. Therefore, this technique can only lead to the computation of fractional phase biases. Following the method described in [9,23], a reference station is chosen and its bias is set to 0. The integer widelane ambiguity and the fractional phase biases of the satellite in view from this station can be determined. It is possible to find another station in the network that shares a common view with some of the previous satellites whose bias has been computed, which leads to the determination of the bias of this station. This process is repeated iteratively. As a consequence, the integer solution of the widelane ambiguity is consistent with the fractional biases and the chosen reference station.

Once the integer widelane ambiguity $N^s_{\text{WL},ij}$ is known, one of the two ambiguities $N^s_i$ or $N^s_j$ must be determined. To do so, choosing the $N^s_i$ ambiguity, the IF combination of the code and phase measurements given in Equation (4) is transformed to:

$$C_{\text{IF},ij} = \rho^s_r + h^s_{r,\text{IF},C_{ij}} + m(E^s)T_Z,$$
$$\lambda_{\text{IF},ij}L_{\text{IF},ij} = \rho^s_r + \lambda_{\text{IF},ij}W + h^s_{r,\text{IF},L_{ij}} + \frac{\gamma_j\lambda_iN_i - \gamma_i\lambda_j(N_i + N^s_{\text{WL},ij})}{\gamma_j - \gamma_i}. \tag{10}$$

Inserting the already determined widelane ambiguity into the right-hand side of the equation leads to:

$$C_{\text{IF},ij} = \rho^s_r + h^s_{r,\text{IF},C_{ij}} + m(E^s)T_Z,$$
$$\lambda_{\text{IF},ij}\tilde{L}_{\text{IF},ij} = \lambda_{\text{IF},ij}L_{\text{IF},ij} + \frac{\gamma_i\lambda_j}{\gamma_j - \gamma_i}N^s_{\text{WL},ij} = \rho^s_r + \lambda_{\text{IF},ij}W + h^s_{r,\text{IF},L_{ij}} + \lambda_{\text{IF},ij}N^s_i. \tag{11}$$

The model of Equation (11) brings the geometric distance together with the troposphere wet delay, the clocks, and the ambiguities. As explained in [9,30], no integer ambiguity can be estimated using this model. Therefore, no integer constraint is added in Equation (11). The determination of the geometrical distance between all the satellites and the ground station receivers, the troposphere delays, the clocks, and the floating ambiguities is performed with a Kalman filter.

In order to recover the integer ambiguities, the estimated geometrical distance is subtracted from the IF phase combination:

$$\lambda_{\text{IF},ij}\hat{L}_{\text{IF},ij} = \lambda_{\text{IF},ij}\tilde{L}_{\text{IF},ij} - \rho^s_r - \lambda_{\text{IF},ij}W = \lambda_{\text{IF},ij}N^s_i + h^s_{r,\text{IF},L_{ij}}. \tag{12}$$

The problem defined by Equation (12) has the same structure as the one defined by Equation (8): an integer ambiguity has to be estimated together with a receiver minus a satellite clock. Hence, the determination of $N_i^s$, $h_{r,\text{IF},L_ij}$, and $h_{\text{IF},L_ij}^s$ follows the same process as the one used for the determination of $N_{\text{WL},ij}^s$, $\mu_{r,ij}$, and $\mu_{ij}^s$.

The code biases can be easily computed together with the ionosphere elongation using the geometry-free (GF) combination of the two code measurements $C_i$ and $C_j$:

$$C_{\text{IF},ij} = \frac{C_i - C_j}{\gamma_i - \gamma_j} = e + \tau_{r,C_{ij}}^s. \tag{13}$$

The individual phase biases are then retrieved from the widelane biases as described in [30]. To do so, it is first mandatory to choose the clock convention used for the definition of the biases on the individual signal. The model given by Equation (1) uses clocks defined as the offset between the satellite and receiver hardware clocks and a reference timescale. For the dissemination of its products, IGS uses its own convention as being the ionosphere-free code clock (see Equation (4)). Therefore, the same convention is used for the definition of the code and phase biases. The code and phase measurements models read thus:

$$\begin{aligned} C_i &= \rho_r^s + h_{r,\text{IF},C_ij} + \gamma_i e + m(E^s)T_Z + b_{r,\text{IF},C_i}^s, \\ \lambda_i L_i &= \rho_r^s + h_{r,\text{IF},C_ij} - \gamma_i e + m(E^s)T_Z + \lambda_i W + \lambda_i N_i^s + b_{r,\text{IF},L_i}^s, \end{aligned} \tag{14}$$

where $b_{r,\text{IF},C_i}^s$ and $b_{r,\text{IF},L_i}^s$ play the role of the undifferenced code and phase biases with respect to the IF clock convention, respectively.

Comparing the code models of Equations (4) and (14) leads to $b_{r,\text{IF},C_i}^s = \gamma_i \tau_{r,C_{ij}}^s$. As $\tau_{r,C_{ij}}^s$ is estimated along with the ionosphere elongation (see Equation (13)), the undifferenced code biases are directly known. It is then possible to define the undifferenced phase bias on the frequency $f_i$ as:

$$b_{r,\text{IF},L_i}^s = \frac{1}{\lambda_i}\left(\tau_{r,L_{ij}}^s + h_{r,\text{IF},L_ij} - h_{r,\text{IF},C_ij}\right). \tag{15}$$

With this definition, the undifferenced phase biases $b_{r,\text{IF},L_i}^s$ and $b_{r,\text{IF},L_j}^s$ must fullfil:

$$\begin{aligned} \mu_{r,ij}^s - \frac{\lambda_i - \lambda_j}{\lambda_i + \lambda_j}\left(\frac{b_{r,\text{IF},C_i}^s}{\lambda_i} + \frac{b_{r,\text{IF},C_j}^s}{\lambda_j}\right) &= b_{r,\text{IF},L_i}^s - b_{r,\text{IF},L_j}^s, \\ \gamma_j \lambda_i b_{r,\text{IF},L_i}^s - \gamma_i \lambda_j b_{r,\text{IF},L_j}^s &= (\gamma_i - \gamma_i)\left(h_{r,\text{IF},L_ij} - h_{r,\text{IF},C_ij}\right). \end{aligned} \tag{16}$$

The inversion of this system of two equations with the two unknowns $b_{r,\text{IF},L_i}^s$ and $b_{r,\text{IF},L_j}^s$ gives the undifferenced phase biases.

This process described for a two-frequencies case can be adapted for a three-frequencies or four-frequencies case (see, for instance, [15,30]). The code and phase measurements on this third or fourth frequency $f_k$ are thus expressed as:

$$\begin{aligned} C_k &= \rho_r^s + \gamma_k e + m(E^s)T_Z + h_{r,\text{IF},C_ij}^s + b_{r,\text{IF},C_k}^s, \\ \lambda_k L_k &= \rho_r^s - \gamma_k e + m(E^s)T_Z + h_{r,\text{IF},C_ij}^s + b_{r,\text{IF},L_k}^s, \end{aligned} \tag{17}$$

with the undifferenced code and phase biases with respect to the IF clock defined by:

$$b_{r,\text{IF},C_k}^s = b_{r,C_k}^s - \frac{\gamma_j b_{r,C_i}^s - \gamma_i b_{r,C_j}^s}{\gamma_j - \gamma_i} \text{ and } b_{r,\text{IF},L_k}^s = b_{r,L_k}^s - \frac{\gamma_j b_{r,C_i}^s - \gamma_i b_{r,C_j}^s}{\gamma_j - \gamma_i}. \tag{18}$$

Adding the new signals on the new frequencies leads to a huge increase in the number of parameters, which is difficult to handle in real time. Indeed, the extra-widelane biases

and ambiguities for the GPS, Galileo, and Beidou constellations have to be tackled on top of the ones usually taken into account in the dual-frequency case. Therefore, the computational load is a big issue.

Particular attention is paid to the production latency of the corrections. The low-frequency varying parameters such as the code biases are estimated in a background thread while high-frequency parameters such as clocks are estimated in a high-priority thread. As of June 2023, the ODTS software processes about 1500 measurements for each constellation at a 5 s interval. The output of the network side of the demonstrator is the routinely computed real-time code and phase biases as well as the real-time satellite orbits and clocks.

### 2.2. Dissemination Standards

The legacy dissemination standard (RTCM) is now replaced in the demonstrator by the new IGS standard, whose messages are defined in a proprietary RTCM message [22]. It defines the corrections of all the constellations in a uniform way and allows the diffusion of the phase biases for integer AR.

The two streams are available at the IGS caster:

- stream SSRA00CNE0 (IGS standard),
- stream SSRA00CNE1 (RTCM standard).

Table 1 synthesizes the various messages available in the output of the ODTS.

**Table 1.** List of real-time messages.

| Constellation | Nature | RTCM Message | IGS Message | Occurrence (s) |
|---|---|---|---|---|
| GPS | orbits/clocks | 1060 | 23 | 5 |
| | code biases | 1059 | 25 | 5 |
| | phase biases (L1, L2, L5), all ambiguities | 1265 | 26 | 5 |
| Glonass | orbits/clocks | 1066 | 43 | 5 |
| | code biases | 1065 | 45 | 5 |
| | phase biases (yaw) | 1266 | 46 | 5 |
| Galileo | orbits/clocks | 1243 | 63 | 5 |
| | code biases | 1242 | 65 | 5 |
| | phase biases (E1, E5a, E5b, E6), all ambiguities | 1267 | 66 | 5 |
| Beidou 2-3 | orbits/clocks | 1261 | 103 | 5 |
| | code biases | 1260 | 105 | 5 |
| | phase biases (B1, B2, B3), widelane ambiguities | 1270 | 106 | 5 |
| | Ionosphere VTEC | 1264 | 201 | 60 |

### 2.3. The User Side of the Demonstrator

2.3.1. Measurement Modeling in the User

The user software is described in [31]. Its main goal is to compute the receiver trajectory. It is based on a Kalman filter that can be run in forward mode for real-time purposes or in forward and backward mode for post-processing. Ambiguity resolution is performed when phase biases are available and compatible. The four main constellations are supported. The software is compatible with the undifferenced multi-frequency bias representation, allowing instantaneous ambiguity resolution when possible and fast convergence. The code is portable and can be embedded in a hand-held device.

The code and phase measurements are modeled with the undifferenced code and phase biases defined with respect to the IGS convention (code ionosphere-free clocks) as

described in Equations (14) and (17). In order to simplify the notations, the $C_{ij}$ subscript of the IF clocks will be removed so that the measurements model reads:

$$C_j = \rho_r^s + \gamma_j e + m(E^s)T_Z + h_{r,\text{IF}}^s + b_{r,\text{IF},C_j}^s,$$
$$\lambda_j L_j = \rho_r^s - \gamma_j e + m(E^s)T_Z + h_{r,\text{IF}}^s + b_{r,\text{IF},L_j}^s + \lambda_j W + \lambda_j N_j^s. \tag{19}$$

The aim of the network side of the demonstrator is to compute and disseminate the satellite part of the code and phase biases appearing in Equation (1) as well as the satellite part of the clock offset: $b_{\text{IF},C_j}^s$, $b_{\text{IF},L_j}^s$, and $h_{\text{IF}}^s$. The user software can retrieve the satellite biases in the RTCM and the IGS standards. The receiver part of these biases has hence to be determined, along with its position. The model defined by Equations (14) and (17) suffers from a rank deficiency. To reduce this deficiency, the measurement equations are reparameterized to let only one receiver clock offset for all the frequencies appear. The code and phase bias definition on all other frequencies is then redefined. Choosing $f_1$ as the reference frequency, it is possible to define a clock relative to the code measurement of the frequency $f_1$ as $\tilde{h}_{r,\text{IF},C_1} = h_{r,\text{IF}} + b_{r,\text{IF},C_1}$ This clock is relative to the frequency $f_1$ in the sense that it embeds both the receiver clock and the receiver code bias of the frequency $f_1$.

With this definition, the code measurement model of the observables on the frequency $f_1$ is rewritten as:

$$\tilde{C}_1 = C_1 + h_{\text{IF}}^s + b_{\text{IF},C_1}^s = \rho_r^s + h_{r,\text{IF}} + b_{r,\text{IF},C_1} + e + m(E^s)T_Z,$$
$$= \rho_r^s + \tilde{h}_{r,\text{IF},C_1} + e + m(E^s)T_Z, \tag{20}$$

$$\lambda_1 \tilde{L}_1 = \lambda_1 L_1 + h_{\text{IF}}^s + b_{\text{IF},L_1}^s = \rho_r^s + h_{r,\text{IF}} + b_{r,\text{IF},L_1} - e + m(E^s)T_Z + \lambda_1 N_1^s + \lambda_1 W,$$
$$= \rho_r^s + \tilde{h}_{r,\text{IF},C_1} + \tilde{b}_{r,\text{IF},L_1} - e + m(E^s)T_Z + \lambda_1 N_1^s + \lambda_1 W. \tag{21}$$

It is then possible to make the code and phase clocks relative to the frequency $f_1$ appear in the code and phase models, respectively, for the other frequencies. Thus, we define new code and phase biases as:

$$\tilde{b}_{r,\text{IF},C_j} = b_{r,\text{IF},C_j} - b_{r,\text{IF},C_1} \quad \text{and} \quad \tilde{b}_{r,\text{IF},L_j} = b_{r,\text{IF},L_j} - b_{r,\text{IF},C_1}, \tag{22}$$

and the measurement model (1) of the code and phase measurements for the frequency $f_{j \neq 1}$ is transformed to:

$$\tilde{C}_j = C_j + h_{\text{IF}}^s + b_{\text{IF},C_j}^s = \rho_r^s + h_{r,\text{IF}} + b_{r,\text{IF},C_j} + \gamma_j e + m(E^s)T_Z,$$
$$= \rho_r^s + \tilde{h}_{r,\text{IF},C_1} + \tilde{b}_{r,\text{IF},C_j} + \gamma_j e + m(E^s)T_Z, \tag{23}$$

$$\lambda_j \tilde{L}_j = \lambda_j L_j + h_{\text{IF}}^s + b_{\text{IF},L_j}^s = \rho_r^s + h_{r,\text{IF}} + b_{r,\text{IF},L_j} - \gamma_j e + m(E^s)T_Z + \lambda_j N_j^s + \lambda_j W,$$
$$= \rho_r^s + \tilde{h}_{r,\text{IF},C_1} + \tilde{b}_{r,\text{IF},L_j} - \gamma_j e + m(E^s)T_Z + \lambda_j N_j^s + \lambda_j W. \tag{24}$$

Since the same clock parameter is shared between all the frequencies, the parameterization proposed by the set of Equations (20), (21), (23) and (24) is called the common clock parameterization. It is performed for each constellation separately, so four receiver clock offsets have to be estimated, one for each constellation.

The method for recovering the integer ambiguities is presented in [31] and recalled here for the sake of completeness. Choosing the frequency $f_1$ as a pivoting frequency, the ambiguity $N_j$ of the phase measurement at the frequency $f_j$ can be expressed as a combination of $N_1$ and the widelane ambiguity $N_{\text{WL},1j} = N_j - N_1$: $N_j = N_1 + N_{\text{WL},1j}$. In the same way, the ambiguity $N_k$ of the phase measurement at the frequency $f_k$ can be expressed using the widelane ambiguity $N_{\text{WL},1j}$ and an extra-widelane ambiguity $N_{\text{EWL},jk}$:

$N_k = N_1 + N_{\text{WL},1j} + N_{\text{EWL},jk}$. The decomposition of the ambiguities on all the single frequencies for the constellations is thus performed as follows:

$$
\text{for GPS}: \begin{cases} N_1 = N_1, \\ N_2 = N_1 + N_{\text{WL},12}, \\ N_5 = N_1 + N_{\text{WL},12} + N_{\text{EWL},25}, \end{cases} \tag{25}
$$

$$
\text{for Galileo}: \begin{cases} N_1 = N_1, \\ N_{5a} = N_1 + N_{\text{WL},15a}, \\ N_{5b} = N_1 + N_{\text{WL},15a} + N_{\text{EWL},5a5b}, \\ N_6 = N_1 + N_{\text{WL},15a} + N_{\text{EWL},5a5b} + +N_{\text{EWL},5b6}, \end{cases} \tag{26}
$$

$$
\text{for Beidou}: \begin{cases} N_{1I} = N_{1I}, \\ N_{3I} = N_{1I} + N_{\text{WL},1I3I}, \\ N_{2I} = N_{1I} + N_{\text{WL},1I3I} + N_{\text{EWL},3I2I}, \\ N_{2b} = N_{1I} + N_{\text{WL},1I3I} + N_{\text{EWL},3I2b}, \\ N_{2a} = N_{1I} + N_{\text{WL},1I3I} + N_{\text{EWL},3I2a}, \\ N_{1C} = N_{1I} + N_{\text{WL},1I1C}. \end{cases} \tag{27}
$$

The widelane $N_{\text{WL},ij}$ and the extra-widelane ambiguities $N_{\text{EWL},jk}$ are estimated from the Melbourne–Wübbena combination [28,29]. It is a linear combination of the carrier-phase widelane and the code narrowlane, whose aim is to build an ionosphere-free geometry-free measurement. In this case, the only remaining quantities are the carrier-phase ambiguity and a combination of the code and phase biases of the involved observables. With the notations of Equation (23), the Melbourne–Wübbena observation model is given by:

$$
\text{MW}(\tilde{L}_i, \tilde{L}_j, \tilde{C}_i, \tilde{C}_j) = \tilde{L}_i - \tilde{L}_j + \frac{\lambda_i - \lambda_j}{\lambda_i + \lambda_j}\left(\frac{\tilde{C}_i}{\lambda_i} + \frac{\tilde{C}_j}{\lambda_j}\right) = N_{\text{WL},ij} + b_{\text{MW},ij}. \tag{28}
$$

With $b_{\text{MW},ij}$, the Melbourne–Wübbena receiver bias given by:

$$
b_{\text{MW},ij} = \frac{\tilde{b}_{r,L_i}}{\lambda_i} - \frac{\tilde{b}_{r,L_j}}{\lambda_j} + \frac{\lambda_i - \lambda_j}{\lambda_i + \lambda_j}\left(\frac{\tilde{b}_{r,C_i}}{\lambda_i} + \frac{\tilde{b}_{r,C_j}}{\lambda_j}\right) \tag{29}
$$

The benefit of obtaining undifferenced OSB lies in the fact that no matter the frequencies $f_i$ and $f_j$ chosen to build the Melbourne–Wübbena combinations, the integer property of the carrier phase ambiguities is always maintained.

The user part of the demonstrator is also able to use the Doppler measurements for the receiver position and velocity estimation. As defined in [1] (Chapter 2), the frequency $f_r^s$ seen by the receiver due to the motion of the GNSS satellite emitting the signal at the frequency $f_j$ is:

$$
f_r^s = \left[\Delta \mathbf{v}_r^s \cdot \frac{\Delta \mathbf{p}_r^s}{||\Delta \mathbf{p}_r^s||} + \dot{h}_r^s\right]\frac{f_j}{c}, \tag{30}
$$

where $\mathbf{v}_r^s$ is the difference between the satellite and the receiver velocities, $\Delta \mathbf{p}_r^s$ is the ranging vector from the receiver to the satellite, $\dot{h}_r^s$ is the clock drift, and $c$ is the velocity of light in vacuum. In the demonstrator, the measurement that is taken into account has a velocity dimension. Therefore, the frequency measured by the receiver is multiplied by the wavelength of the original signal, so the Doppler observable reads:

$$
D_j = \lambda_j f_r^s = \Delta \mathbf{v}_r^s \cdot \frac{\Delta \mathbf{p}_r^s}{||\Delta \mathbf{p}_r^s||} + \dot{h}_r^s. \tag{31}
$$

As it can be seen in Equation (31), the velocity state of the receiver is involved in the Doppler model. As the receiver trajectory estimation is performed by means of a Kalman filter, the link with the position state is done in the prediction part of the filter. As described in references [32,33], using the Doppler measurement improves the positioning accuracy. This will be assessed in the following sections.

The state vector to be estimated consists of the receiver position, the receiver velocity, the receiver clock relative to $P1$ for GPS, the receiver clock relative to $E1$ for Galileo, the receiver clock relative to $B1I$ for Beidou, the ionospheric elongation, the wet part of the zenithal tropospheric delay, the code and phase biases on all the frequencies except the $f_1$ one, as well as the $N_1$, $N_{WL,ij}$, and $N_{EWL,jk}$ ambiguities. The estimation is performed by a Kalman filter, implemented in a square root formulation, in the so-called $UDU^T$ form (see references [34] (Chapter 5), [35], or [36], for instance). The integer ambiguities are recovered with a bootstrap method [37].

After having solved for the widelane and extra-widelane ambiguities, the ambiguities on each carrier-phase measurement are recovered thanks to the cascading scheme given by the Equations (25)–(27). Nevertheless, the integer $N_1$ ambiguity is still to be solved for each constellation. This can be performed with the Best Integer Equivariant (BIE) method, for instance [38].

### 2.3.2. Instantaneous Ambiguity Resolution

Undifferenced ambiguity resolution is now a well-known concept, at least in the dual-frequency GPS/Galileo context, and its performance is well established (see references [9–11,39], for instance). Nevertheless, it suffers from an important drawback: the convergence time, which is intrinsic to the use of a Kalman filter to estimate the ambiguities and can be, in this case, in the order of tens of minutes.

Multi-frequency ambiguity resolution is relatively new, in particular with the use of more recent constellations such as Galileo or Beidou. Recently, a new technique called OEUFS (Optimal Estimation using Uncombined Four-frequency Signals) has been proposed [30]. The idea is to form a consistent set of undifferenced and uncombined phase biases that can be processed optimally at the user level. Depending on the constellations and biases used, instantaneous widelane ambiguity resolution or even full ambiguity resolution can be achieved, leading to instantaneous centimeter-accuracy positioning. We recall here the main ingredients of the method and propose a way forward to further improve its robustness.

First, we make a noise analysis for a single satellite. We assume that we can solve for all the widelane ambiguities, and we compute the noise of all the widelanes and extra-widelanes for each constellation, as done in [16,40]. From this analysis, it is possible to select the extra-widelanes with the smallest noise in order to be able to fix their ambiguities. From this point, it is possible to add these ambiguities as constraints for the receiver position estimation (see [31]). Table 2 gives the expected noises of the receiver position using such noise-optimal extra-widelane fixed ambiguities. Using Galileo and Beidou is particularly interesting with an expected range noise of around 20 cm. We assume here a noise of 0.5 m for the code and 0.003 m for the phase.

**Table 2.** Single satellite noise analysis.

| Constellation | Ranging Noise of Optimal Widelanes Combination (cm) |
| --- | --- |
| GPS (L1, L2, L5) | 30 |
| GAL (E1, E5a, E5b, E6) | 19 |
| BEI-2 (B1I, B2I, B3I) | 33 |
| BEI-3 (B1I, B1C, B2a, B2b, B3I) | 20 |

Second, we work at the user level and we assume that the code and phase biases are available for each signal. The position of the user is estimated along with the phase ambiguities, and we obtain the following results:

1. all the widelanes can easily be solved;
2. the obtained solution is accurate enough to enter the narrowlane convergence domain, leading to centimeter accuracy.

If the ambiguity resolution for the widelanes step can easily be achieved using any method, a partial ambiguity resolution method such as the BIE is preferred to solve for the narrowlane (but can also be applied to solve for the widelanes). The entire process is summarized in Figure 3.

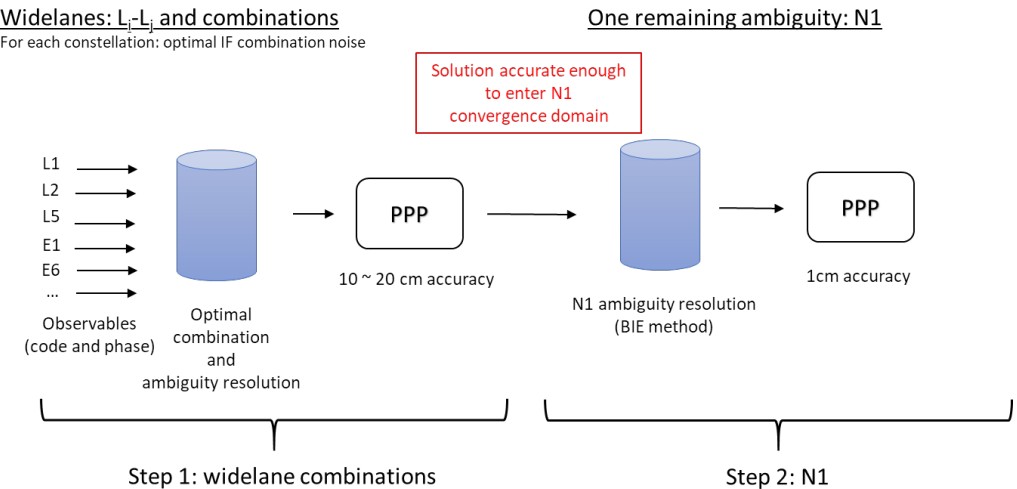

**Figure 3.** Instantaneous ambiguity resolution diagram. The optimal combination is to be understood as the extra-widelane combination leading to the smallest noise possible.

### 2.3.3. The PPP-Snapshot Concept

On the user side, the OEUFS method has been implemented in the so-called PPP-Snapshot mode, taking its name from the fact that the ambiguity is reinitialized at each epoch and the Kalman filter does not exploit the fact that the carrier-phase ambiguity is constant over a single passage. AR is thus performed at each epoch without any knowledge of the previously resolved integer ambiguity. As shown in [41], the convergence of the receiver position is mainly driven by the number of frequencies used for AR. Hence, the benefit of using this configuration lies in the fact that the long convergence time due to the phase ambiguity parameter estimation in the filter disappears, and there is no need to detect or repair the cycle slips, as they are not estimated at each epoch any more. The other parameters of the Kalman filter, such as the ionosphere elongation, the troposphere delay, or the dynamical model, have to be estimated in the classical way by the filter. The ionospheric elongation can thus be maintained by the filter over the time, even in case of a gap in the measurements. Doing so greatly improves the ability of the Snapshot mode to estimate the ambiguities again without convergence at the end of the measurement gap.

However, as a main drawback, this method suffers from the difficulty of solving for the narrowlane ambiguity, which leads to a possible degradation of the accuracy. On top of that, the compatible phase combinations are limited to Galileo E6 and Beidou-3 signals. The number of real-time compatible stations at the IGS is limited but increases over time, as seen in Figure 4.

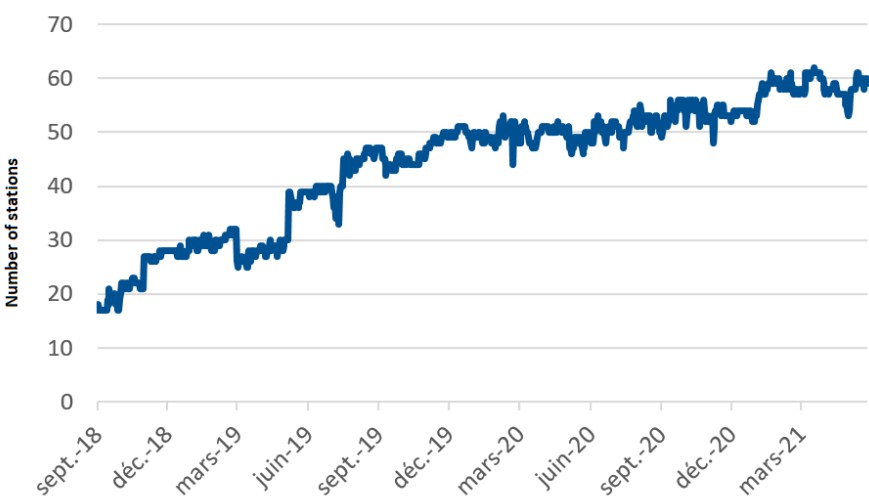

**Figure 4.** Number of E6-compatible stations at the IGS.

### 2.4. Computation of the post-processed Phase Biases

To overcome the different limitations of the real-time phase bias productions (lack of compatible stations, RTCM standard not up to date for some BDS-3 real-time observables), it has been decided to implement a routine production of post-processed phase biases besides the real-time ones. The idea is to compute the phase biases in addition to the already existing products at the IGS. GFZ rapid products were chosen because they support all the GNSS constellations and are of low latency (a few days). The process is described in Figure 5.

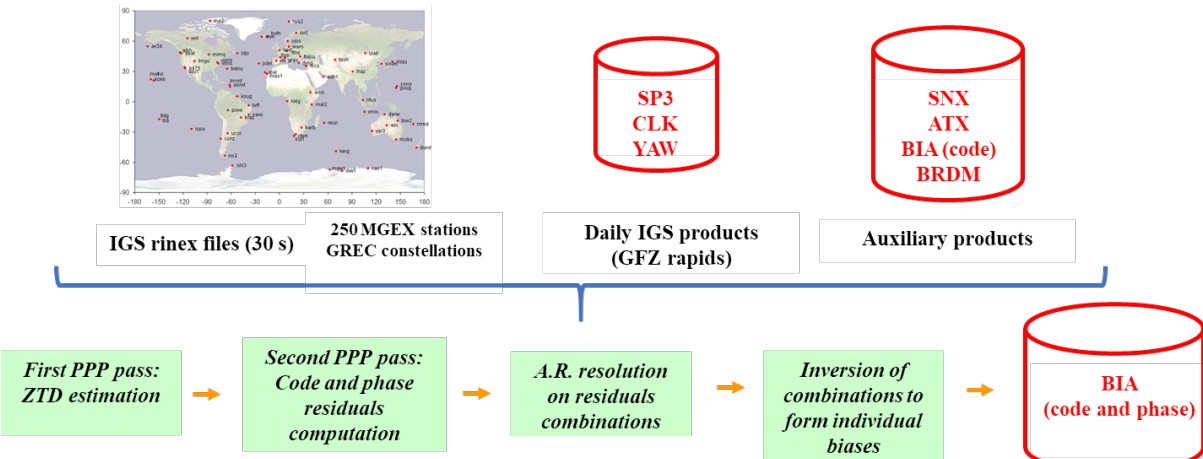

**Figure 5.** Phase biases computation diagram.

It is first mandatory to collect all the necessary data (rinex files of a network of more than 260 IGS stations, SP3, CLK and YAW files, and auxiliary products such as the sinex file of the day, antex files, code biases, and broadcasts). The phase biases can then be computed following these steps:

1. a first PPP pass (forward, backward) to compute the troposphere elongation of the stations;
2. the computation of the observed minus computed (OmC) values of all the code and phase observables;
3. the resolution of ambiguities on a chosen set of the OmC values combinations (the output of this process is the satellite fractional phase biases of the combinations) in the same way as the one done for the production of the real-time products;

4.     the inversion of the combined fractional phase biases to recover individual phase biases.

For step 1, the IF code and phase measurements are used for the correction part of the Kalman filter. Since the satellite code biases and clock offsets are known from the retrieved data, and since the stations receiver positions are known from the weekly combination of the IGS daily combined sinex solutions, the only unknowns that remain in the modeled observables are the wet zenithal troposphere delay, the receiver IF code and phase biases, as well as the IF float ambiguity:

$$
\begin{aligned}
C_{\mathrm{IF},ij} + h_{\mathrm{IF}}^s - \rho_{r,\mathrm{IF}}^s - b_{\mathrm{IF},C_{\mathrm{IF},ij}}^s &= m(E^s)T_Z + \tilde{b}_{r,\mathrm{IF},C_{\mathrm{IF},ij}}, \\
L_{\mathrm{IF},ij} + h_{\mathrm{IF}}^s - \rho_{r,\mathrm{IF}}^s &= m(E^s)T_Z + \tilde{b}_{r,\mathrm{IF},L_{\mathrm{IF},ij}} + \lambda_{IF,ij}N_{\mathrm{IF},ij},
\end{aligned}
\tag{32}
$$

with $b_{\mathrm{IF},C_{\mathrm{IF},ij}}^s$, $\tilde{b}_{r,\mathrm{IF},C_{\mathrm{IF},ij}}$, $b_{\mathrm{IF},L_{\mathrm{IF},ij}}^s$, $\tilde{b}_{r,\mathrm{IF},L_{\mathrm{IF},ij}}$, $\lambda_{IF,ij}$, and $N_{\mathrm{IF},ij}$ being the satellite/receiver code IF biases, the satellite/receiver phase IF biases, the IF wavelength, and the IF ambiguity, respectively. Since the IF ambiguity is defined by:

$$
N_{IF,ij} = \frac{\gamma_j \lambda_i N_i - \gamma_i \lambda_j N_j}{\gamma_i - \gamma_j},
\tag{33}
$$

its integer property is not maintained. Therefore, it can be estimated along with the IF receiver phase bias. Defining $\hat{b}_{\mathrm{IF},L_{\mathrm{IF},ij}} = \tilde{b}_{r,\mathrm{IF},L_{\mathrm{IF},ij}} + \lambda_{IF,ij}N_{\mathrm{IF},ij}$, the model of the code and phase measurements used to estimate the wet zenithal troposphere delay is:

$$
\begin{aligned}
C_{\mathrm{IF},ij} + h_{\mathrm{IF}}^s - \rho_{r,\mathrm{IF}}^s - b_{\mathrm{IF},C_{\mathrm{IF},ij}}^s &= m(E^s)T_Z + \tilde{b}_{r,\mathrm{IF},C_{\mathrm{IF},ij}}, \\
L_{\mathrm{IF},ij} + h_{\mathrm{IF}}^s - \rho_r^s &= m(E^s)T_Z + \hat{b}_{r,\mathrm{IF},L_{\mathrm{IF},ij}}.
\end{aligned}
\tag{34}
$$

The OmC values computed for step 2 consist of the difference between the raw code and phase measurements, from which are subtracted the geometric distance corrected by the phase center offset and the phase center variations, the Sagnac and Shapiro effects, the wind-up term for the phase measurement, the satellite clock offsets, the satellite code biases, and the previously computed wet zenithal troposphere.

The process for the determination of the satellite phase biases at step 3 is the same as the one presented in Section 2.1 for the computation of the real-time phase biases. The problem has been made simpler by the previous determination of the troposphere delay and the use of the ground station positions in the sinex file. In addition to this, since this process is not performed in real-time, the entire passages of the satellites over the ground stations can be considered to retrieve the integer nature of the ambiguities. Indeed, once the cycle slips have been corrected, the ambiguities must be constant over the whole visibility interval from each satellite to each station. This constraint enhances the precision of the phase-bias determination

Step 4 uses a technique similar to the one presented in [16]. Using this process, the computed phase biases are compatible with the OEUFS method. Table 3 shows the different combinations of OmC values used for each constellation. The final product is combined with the code biases to form a BIA file, which is uploaded to the website of the project (http://www.ppp-wizard.net/daily.html, accessed on 12 July 2023). On this webpage, both the real-time and the post-processed phase biases are available.

In the following sections of this article, the products computed with the real-time architecture will be denoted "RT products", whereas the ones computed with the exposed post-process procedure will be called "POST products".

**Table 3.** Choices of combinations for ambiguity resolution. MW WL stands for Melbourn–Wübbena widelane, IF stands for ionosphere-free phase, and WL IF stands for ionosphere-free phase widelanes combinations.

|      | MW WL   | IF      | WL IF      | WL IF       | WL IF       |
|------|---------|---------|------------|-------------|-------------|
| GPS  | F1/F2   | F1 / F2 | F1/F2/F5   | –           | –           |
| GAL  | E1/E5a  | E1/E5a  | E1/E5a/E6  | E1/E5a/E5b  | –           |
| BEI  | B1I/B3I | B1I/B3I | B1I/B3I/B2I| B1I/B3I/B1c | B1I/B3I/B2a |

## 3. Phase-Bias Results Assessment

### 3.1. Assessment of the CNES Phase Biases

In order to highlight the differences between the RT and the POST products, this section displays the time series for several products for the four main constellations over five days, ranging from Day of Year (DOY) 2022-079 to DOY 2022-083. Table 4 gives the observable biases that can be found in the files available on the PPP-WIZARD project webpage for the proposed days. These observables are separated by constellations and frequency bands. They are given as per their rinex codes with constellation letter, frequency number, and tracking mode. Figures 6–8 display the code and phase biases computed by the PPP-WIZARD demonstrator and made available for the scientific community. Note that no phase biases are produced for Glonass. The Russian constellation uses Frequency Division Multiple Access (FDMA) to distinguish the signals emitted by the satellites. This prevents us from solving for the integer ambiguities of Glonass carrier-phase observables. The graphs shown are limited to some frequencies for brevity.

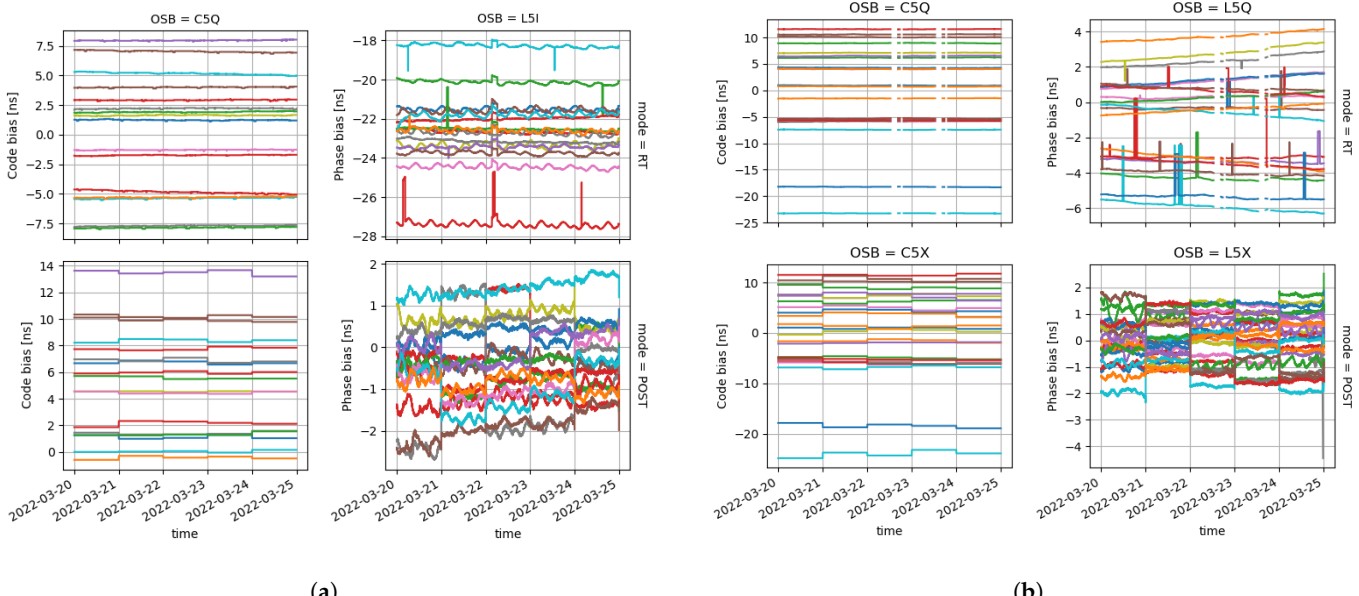

**Figure 6.** (**a**) Code- and phase-biases comparison for GPS C5Q and L5I signals (each color represents a satellite). (**b**) Code- and phase-biases comparison for Galileo L5 band (each color represents a satellite).

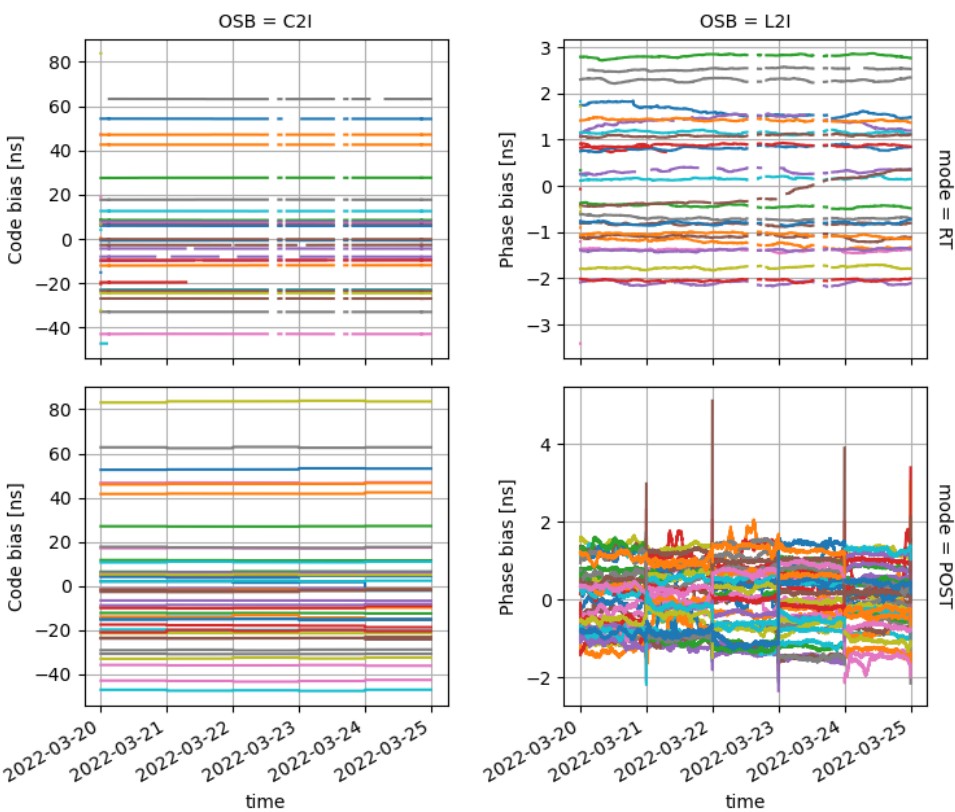

**Figure 7.** Code- and phase-biases comparison for Beidou C2I and L2I signals (each color represents a satellite).

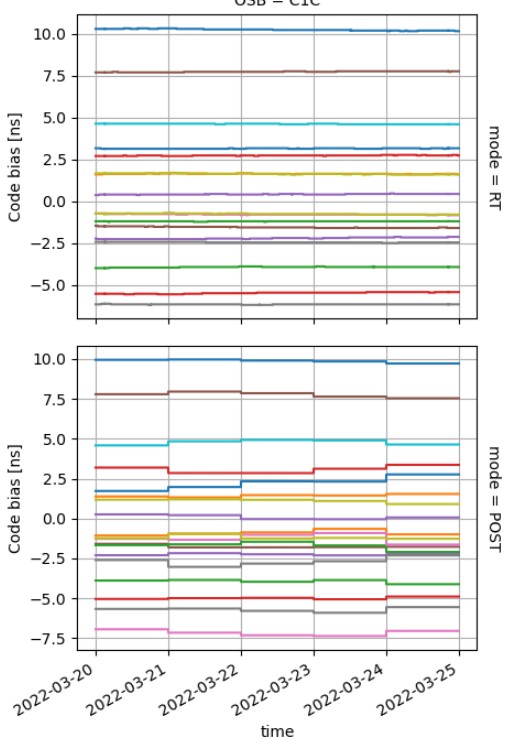

**Figure 8.** Code-biases comparison for Glonass C1C frequency (each color represents a satellite).

**Table 4.** Code and phase biases computed by the PPP-WIZARD software for the DOY 2023-079 to 2023-083.

| System | Frequency | Code Bias | | Phase Bias | |
|---|---|---|---|---|---|
| | | RT | POST | RT | POST |
| GPS | L1 | C1C, C1P, C1W | C1C, C1W | L1C | L1C |
| | L2 | C2S, C2L, C2X | C2C, C2S, C2L, C2W | L2W | L2W |
| | L5 | C5Q, C5X | C5Q, C5X | L5I | L5I |
| GAL | E1 | C1C | C1X | L1C | L1X |
| | E5a | C5Q | C5X | L5Q | L58X |
| | E5b | C7Q | C7X | L7Q | L7X |
| | E6 | C6C | C6X | L6C | L6X |
| BEI | B1C | C1P | C1P, C1X | – | L1X |
| | B1I | C2I | C2I | L2I | L2I |
| | B2a | C5P | C5X | – | L5X |
| | B3A | C6I | C6I | L6I | L6I |
| | B2b | C7I | C7I, C7Z | L7I | L7I |
| GLO | G1 | C1C | C1C | – | – |
| | G2 | C2C | C2C | – | – |

The first row of these figures depicts code or phase biases computed in real-time mode, whereas the second row contains the biases produced with the previously described post-processed mode. The post-processed code and phase biases for all the presented constellations and signals have discontinuities at the break of each day. This is explained by the fact that the products used every day for the POST bias calculations are discontinuous themselves. Such daily jumps in the time series are not to be seen in the RT products. However, some outliers are visible in the RT phase biases for all the constellations. The RT code biases are slowly drifting over the five days duration, whereas the POST code biases are constant over a whole day. Missing values can be seen in the RT products for Galileo and Beidou. This phenomenon may be due to some latency in obtaining the GNSS observables from all the stations.

### 3.2. Comparison of the CNES Phase Biases

The aim of this section is to compare the RT and POST code and phase Observable Specific Biases (OSB) produced by CNES with the ones computed by the School of Geodesy and Geomatics (SGG) at Wuhan University [42]. A focus is put on the stability of the difference between the CNES and the SGG biases over time. To do so, a first mean adjustment is performed. The mean value of the biases over all the satellites of a constellation and for each production center (CNES or SGG) at every epoch is subtracted from the biases. The OSB comparison is then performed with the mean-adjusted biases:

$$\delta_{\text{OSB}}(t,s) = \left[ \text{OSB}_{\text{CNES}}(t,s) - \overline{\text{OSB}}^s_{\text{CNES}}(t) \right] - \left[ \text{OSB}_{\text{SGG}}(t,s) - \overline{\text{OSB}}^s_{\text{SGG}}(t) \right], \quad (35)$$

where $\text{OSB}(t,s)$ represents any OSB product at epoch $t$ for the satellite $s$ and $\overline{\text{OSB}}^s(t)$ is the mean over all the satellites $s$ at epoch $t$. Removing the mean value per constellation would have no impact from a user perspective, since this mean common term would be hidden in the clock parameter $\tilde{h}^s_r$ defined for each constellation in Equation (23) and common to all signals emitted by the satellites of this constellation. Note that the same mean adjustment is performed for the evaluation of the Galileo High Accuracy Service (see [43,44], for instance).

To evaluate the stability of the difference between the adjusted biases between both centers, the empirical standard deviation has to be computed. However, since the CNES POST and the SGG products have discontinuities at the end of each day, the standard

deviation is computed over a one-day period. Then, the median of the five standard deviations for each satellite and each signal is computed and displayed in Figures 9–12.

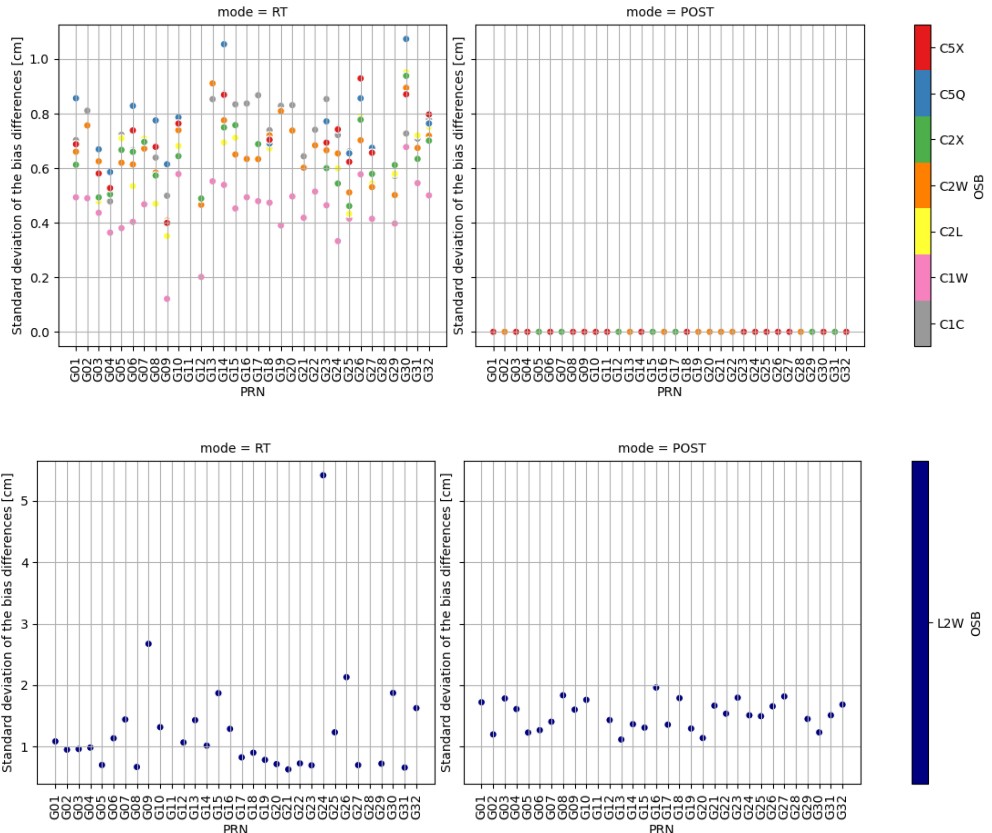

**Figure 9.** Median of the standard deviations of the difference between the CNES RT/POST biases and SGG biases for the GPS constellation. Each standard deviation is computed over a period of one day. The left column depicts the RT products and the right one depicts the POST products. The first row is dedicated to the code biases and the second one to the phase biases.

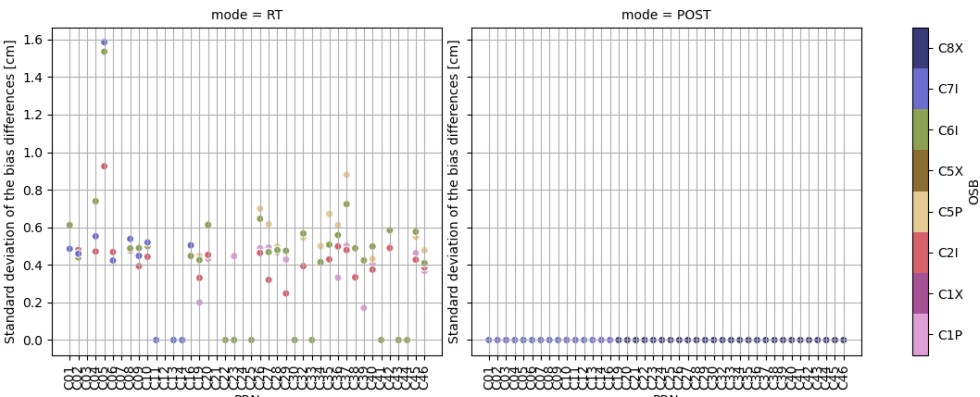

**Figure 10.** *Cont.*

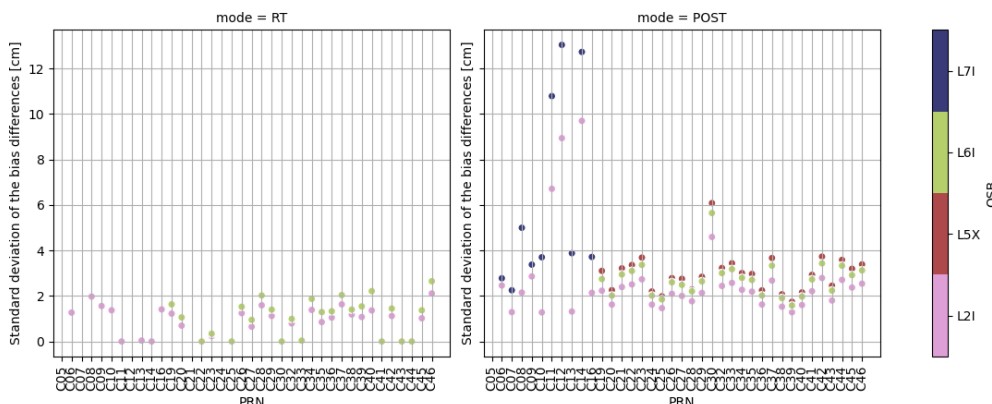

**Figure 10.** Median of the standard deviations of the difference between the CNES RT/POST biases and SGG biases for the Beidou constellation. Each standard deviation is computed over a period of one day. The left column depicts the RT products and the right one depicts the POST products. The first row is dedicated to the code biases and the second one to the phase biases.

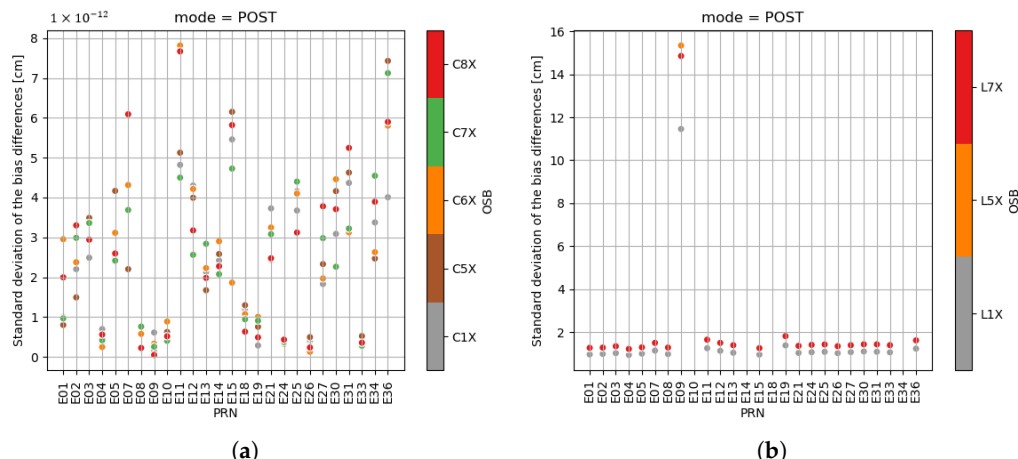

(**a**)              (**b**)

**Figure 11.** Median of the standard deviations of the difference between the CNES POST biases and SGG biases for the Galileo constellation. Each standard deviation is computed over a period of one day. The left figure depicts the code products and the right one depicts the phase products. (**a**) Code biases. (**b**) Phase biases.

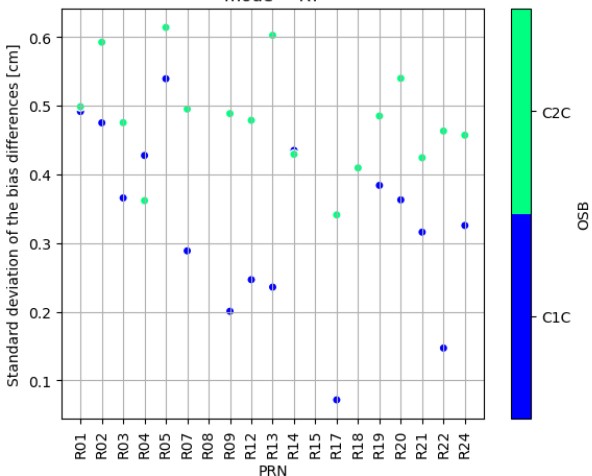

**Figure 12.** Median of the standard deviations of the difference between the CNES RT biases and SGG biases for the Glonass constellation. Each standard deviation is computed over a period of one day.

The median of the standard deviations of the differences between CNES and SGG is expressed in units of distance (centimeters) to evaluate the impact on the pseudorange and carrier-phase measurements. In Figures 9 and 10, the left column is the difference between the CNES RT products and SGG ones, whereas the right column is for the comparison between CNES POST and SGG products. Since the real-time OSB computed by CNES are not in the set of the OSB computed by SGG for Galileo, Figure 11 displays only the stability comparison for the POST products. For Glonass, the RT code biases could only be computed. On the graphs, each color represents a given signal. Some satellites may have missing values if they are not found in either the CNES RT, the CNES POST, or the SGG products.

Since the CNES POST and the SGG code biases are constant, the mean-adjusted values are also constant, and their standard deviations are zero. This is not the case for the RT products. It has to be noted that the standard deviations of the difference are smaller for the code biases than for the phase biases, meaning that the latter is less stable than the former, which is in accordance with the conclusions drawn by [42]. For GPS, the POST phase bias difference with the SGG ones is more stable than their RT counterparts. For Beidou, this is the case only for the Beidou-3 satellites. For Beidou-2 ones, the POST phase biases are less stable than the RT ones. For Galileo, extremal values are reached for the E09 satellite. This is caused by a sudden decrease in the phase bias value at the end of the time series for this spacecraft. The post-processed standard deviation over five days is around 2 cm for GPS and Galileo (except for the extreme case described above) and 4 cm for the Beidou-3 satellites.

## 4. Precise Point Positioning Results

### 4.1. CNES Evaluation

The first evaluation to be done is to demonstrate the benefit of ambiguity fixing. This analysis has been performed on the BRST station located in the north of France with a measurement sampling rate of 30 s for 45 min. Figure 13a shows the horizontal positioning errors in float mode (without the biases), in full GNSS mode. Here, the main driver of the error is the noise of the code. The low code noise of the Galileo measurements allows a positioning at a 20 cm level. Using the phase biases with instantaneous widelane ambiguity resolution leads to a horizontal static positioning accuracy of 10 cm, as depicted in Figure 13b. When the last ambiguity ($N1$) is solved, it is possible to obtain sub-centimeter accuracy instantaneously (Figure 13c). Note that this last resolution requires very accurate phase biases and clean measurements (open sky stations). Table 5 summarizes all these results for the positioning errors for the three cases in the East and North directions.

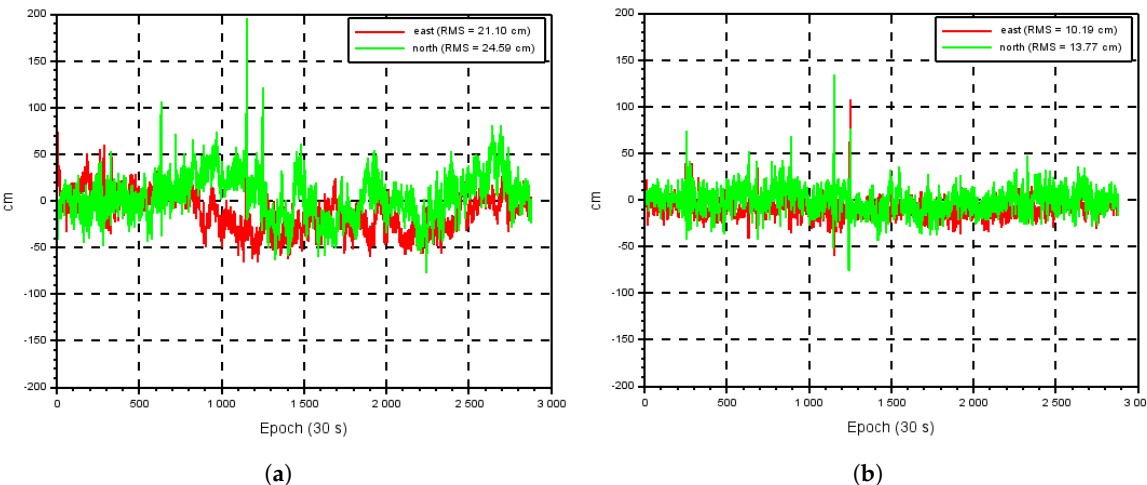

(**a**)                    (**b**)

**Figure 13.** *Cont.*

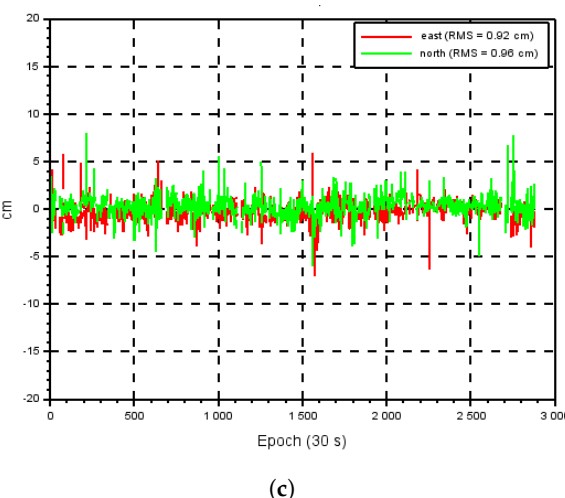

(c)

**Figure 13.** Difference in the positioning of the static station BRST with no ambiguity resolution, the widelane ambiguity resolution only, and resolution of all the ambiguities. (**a**) Standard PVT. (**b**) PVT with instantaneous widelane ambiguity resolution. (**c**) PVT with instantaneous ambiguity resolution, all ambiguities.

**Table 5.** Results accuracy for three positioning modes for the BRST station, with measures processed at a time interval of 30 s.

|  | PVT | WL Only | All Ambiguities |
|---|---|---|---|
| RMS East error [cm] | 21.10 | 10.19 | 0.92 |
| RMS North error [cm] | 24.59 | 13.77 | 0.96 |

In order to assess the better precision of the POST products with respect to the RT ones, a PPP solution is computed with three PPP modes: the PPP-float mode, the PPP-AR mode, and the new PPP-Snapshot mode presented in Section 2.3. The receiver position and velocity are estimated in the Kalman filter along with the receiver clock offsets for each constellation, the receiver code and phase biases for all the frequencies, the carrier phase combinations ambiguities for all the satellites, the ionosphere elongation, and the troposphere delay. A simple linear dynamical model is used in the propagation phase of the Kalman filter to link the receiver position and velocity. Its position and velocity at time $t_k$ are denoted $p_k$ and $v_k$, respectively. This dynamical model reads:

$$
\begin{aligned}
p_{k+1} &= p_k + v_k \Delta t + \omega_{p,k}, \\
v_{k+1} &= v_k + \omega_{v,k},
\end{aligned}
\tag{36}
$$

where $\Delta t = t_{k+1} - t_k$ is the time interval between two steps of the Kalman filter and $\omega_{p,k}$ and $\omega_{v,k}$ are the centered white process noises for the position and velocity, respectively, whose standard deviations $\sigma_{\omega_p}$ and $\sigma_{\omega_v}$ are to be chosen as parameterization of the filter, as well as the a priori standard deviation $\sigma_{p_0}$ and $\sigma_{v_0}$. Two cases are considered for this study:

1.  in the first case, called static (STA), the velocity process noise standard deviation $\sigma_{\omega_v}$ and the velocity a priori standard deviation $\sigma_{v_0}$ are set to zero, in such a way that the velocity is not estimated by the filter;

2.  in the second case, called kinematic (KIN), the velocity a priori and process noise standard deviations are not zero, and the position process noise standard deviation is set to zero.

For the other estimated parameters, the transition matrix is the identity matrix. Table 6 describes the a priori and process noise standard deviations that have been chosen. The computations have been done for the TLSG station for the day 2023-073 (14 March) with

a data sampling rate of 1 Hz. The results in terms of mean, standard deviation, and 95 percentile of the position and velocity errors are given in Table 7. Note that, as presented in [40], the statistics are computed after 30 min of convergence.

**Table 6.** Kalman filter settings for the static (STA) and kinematic (KIN) modes.

| | | |
|---|---|---|
| GPS observables | | Code, carrier phase, Doppler shift |
| Galileo observables | | Code, carrier phase, Doppler shift |
| Beidou observables | | Code, carrier phase, Doppler shift |
| Glonass observables | | Code, Doppler shift |
| Iono a priori standard deviation [m] | | 10 |
| Iono process noise standard deviation [m] | | 0.025 |
| Tropo model | | Saastamoinen |
| Wet tropo a priori standard deviation [m] | | 0.5 |
| Wet tropo model standard deviation [m] | | $5 \times 10^{-6}$ |
| Galileo code measurements standard deviation [m] | | 0.5 |
| GPS code measurements standard deviation [m] | | 1 |
| Beidou code measurements standard deviation [m] | | 10 |
| Carrier phase measurements standard deviation [cm] | | 5 |
| Doppler shift measurements standard deviation [cm/s] | | 5 |
| | **STA** | **KIN** |
| Position a priori standard deviation [m] | 10.0 | 10.0 |
| Position process noise standard deviation [m] | 1.0 | 0.0 |
| Velocity a priori standard deviation [m/s] | 0.0 | 1.0 |
| Velocity process noise standard deviation [m/s] | 0.0 | 0.025 horizontal 0.05 vertical |

**Table 7.** Positioning errors for the TLSG station with the two different products computed by the WIZARD demonstrator (RT or POST), with three positioning modes (PPP-float, PPP-AR, or PPP-Snapshot) in the static (STA) or kinematic (KIN) dynamic settings.

| | | | **Horizontal** | | | **Vertical** | | |
|---|---|---|---|---|---|---|---|---|
| **Real-Time Products** | | | | | | | | |
| | | | **mean** | **std.** | **95%** | **mean** | **std.** | **95%** |
| PPP-float | KIN | Pos. error [cm] | 42.41 | 21.61 | 80.84 | −37.15 | 53.71 | 49.62 |
| | | Vel. error [cm/s] | 0.48 | 0.27 | 0.97 | −0.02 | 0.81 | 1.32 |
| | STA | Pos. error [cm] | 41.14 | 20.56 | 77.38 | −36.53 | 48.23 | 39.00 |
| PPP-AR | KIN | Pos. error [cm] | 1.38 | 3.67 | 2.39 | −0.30 | 4.94 | 5.95 |
| | | Vel. error [cm/s] | 1.08 | 1.32 | 2.07 | 0.60 | 2.13 | 3.78 |
| | STA | Pos. error [cm] | 1.29 | 1.72 | 2.42 | −0.47 | 4.20 | 5.84 |
| PPP-Snapshot | KIN | Pos. error [cm] | 40.07 | 18.94 | 72.44 | −13.75 | 53.68 | 68.91 |
| | | Vel. error [cm/s] | 0.77 | 0.68 | 2.42 | 0.12 | 1.24 | 2.12 |
| | STA | Pos. error [cm] | 43.37 | 20.64 | 80.44 | −11.93 | 60.70 | 76.01 |
| **Post-Processed Products** | | | | | | | | |
| | | | **mean** | **std.** | **95%** | **mean** | **std.** | **95%** |
| PPP-float | KIN | Pos. error [cm] | 35.35 | 18.80 | 70.52 | −8.65 | 48.62 | 66.49 |
| | | Vel. error [cm/s] | 0.44 | 0.24 | 0.88 | −0.02 | 0.75 | 1.21 |
| | STA | Pos. error [cm] | 34.29 | 18.07 | 67.66 | −8.11 | 43.83 | 59.34 |

**Table 7.** *Cont.*

| | | | Post-Processed Products | | | | | |
|---|---|---|---|---|---|---|---|---|
| | | | Horizontal | | | Vertical | | |
| | | | **mean** | **std.** | **95%** | **mean** | **std.** | **95%** |
| PPP-AR | KIN | Pos. error [cm] | 0.86 | 0.59 | 1.90 | −0.40 | 2.83 | 4.52 |
| | | Vel. error [cm/s] | 0.64 | 0.51 | 1.45 | −0.99 | 1.17 | 0.94 |
| | STA | Pos. error [cm] | 0.87 | 0.60 | 1.90 | −0.26 | 2.81 | 4.67 |
| PPP-Snapshot | KIN | Pos. error [cm] | 33.19 | 16.29 | 62.76 | 16.86 | 46.09 | 87.68 |
| | | Vel. error [cm/s] | 1.08 | 0.58 | 2.15 | −1.46 | 1.91 | 1.96 |
| | STA | Pos. error [cm] | 34.42 | 17.03 | 65.91 | 40.09 | 50.74 | 121.15 |

When comparing the results between the first table with the RT products and the second table with the POST products, the statistics obtained with the POST products are better than the ones obtained with the real-time products, except for the PPP-Snapshot mode in the vertical direction, for which the mean and the 95 percentile of the kinematic position are both increased with the POST products. As demonstrated with these results, the use of the routinely computed new post-processed phase biases leads to an increase of the positioning accuracy. As excepted, the position solution with ambiguity resolution is more precise by an order of magnitude than the solution in float mode, with an almost 80% relative improvement in kinematic and static modes. The standard deviation and the 95 percentile are also decreasing. However, the ambiguity resolution degrades the velocity error in the kinematic mode. The PPP-Snapshot mode plays the role of an intermediate mode between the PPP-float and the PPP-AR modes in terms of accuracy. The position results with the PPP-Snapshot mode are better than the ones for the PPP-float mode but still worse than the PPP-AR mode. This is explained by the fact that the ambiguity is reinitialized at each epoch, and the Kalman filter does not exploit the fact that the carrier-phase ambiguity is constant over a single passage. This allows a positioning without convergence, but at the price of less precise results. This accuracy could be improved by using external atmospheric corrections.

In terms of convergence time, a special computation case with RT and POST products has been performed with a reinitialization of the filter every two hours. This leads to twelve convergence phases for a day, which can be sufficient to obtain a representative convergence time. Table 8 summarizes the time needed to have a permanent positioning error under 5 cm for the horizontal component and below 10 cm for the vertical one. The convergence times are always smaller using POST products than RT ones. In addition to this, the kinematic mode leads to longer convergence times than the static mode.

**Table 8.** Convergence time for the positioning of the TLSG station on 14 March 2023, with the RT or the POST products, in static (STA) or kinematic (KIN) modes.

| | Horizontal Convergence [min] | | Vertical Convergence [min] | |
|---|---|---|---|---|
| | **STA Mode** | **KIN Mode** | **STA Mode** | **KIN Mode** |
| RT products | 54 | 55 | 72 | 73 |
| POST products | 25 | 31 | 25 | 29 |

The benefit of using the Doppler shift measurement on the user side of the PPP-WIZARD demonstrator is also assessed in this section. To do so, the Doppler shift model described in Section 2.3 has been implemented and tested on the measurements collected on a car campaign held in the city center and the surroundings of Toulouse, France, in April 2022. A car has been equipped with a Novatel antenna on the roof connected to a signal splitter. The antenna signal is directed on one hand to a ProPack 6 receiver and a Novatel Inertial Measurement Unit (IMU) in order to compute a reference trajectory, and on the other hand to a Septentrio PolaRx5 receiver, whose data have been used in this section in

order to estimate the car position. The lever arms from the IMU to the antenna have been calibrated to a centimeter level. The experimental setting is depicted in Figure 14a and the path traveled by the car in Toulouse city in Figure 14b. Despite what is commonly done in the navigation community, the measures have been sampled at 10 Hz rather than 1 Hz.

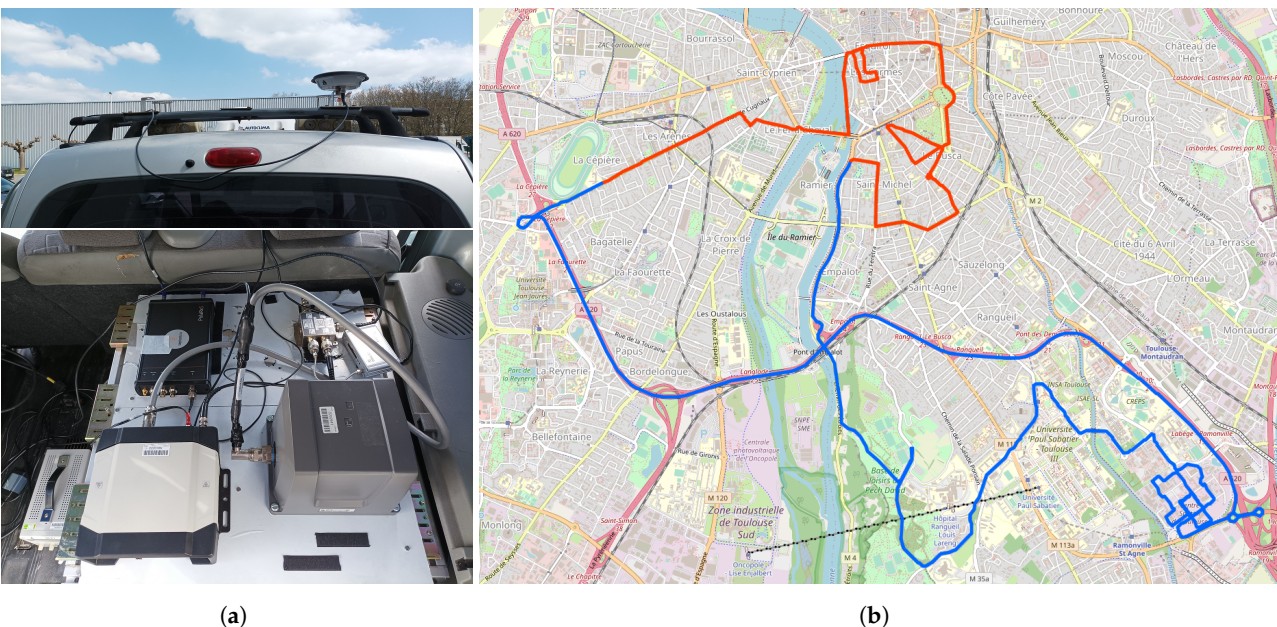

(**a**)                                          (**b**)

**Figure 14.** Experimental setting and path traveled during the car campain. (**a**) Experimental setting. (**b**) Reference trajectory. In blue the "sub-urban" part, in red the "urban" part.

The trajectory of the car has been estimated with the user side of the demonstrator using the code, phase, and Doppler measurements from the GPS and Galileo satellites. The real-time and post-processed products have been used to correct the satellite broadcast ephemeris and clocks, as well as to add code and phase biases to the undifferenced and uncombined measurement model given in Equation (1). The comparison in terms of mean, standard deviation, and 95 percentile of the horizontal errors between the estimated trajectory and the reference is presented in Table 9. When comparing the errors using only the code and phase measurements on one hand and the three types of measurements on the other hand, the accuracy of the position is better when the measures are sampled at 10 Hz rather than at 1 Hz. With the POST products, the mean horizontal position accuracy is improved by 28% and the RMS by 43% or 56% in the Snapshot and full AR modes, respectively, with the 10 Hz Doppler measurement. Conversely, the use of the 1 Hz measurements does not improve the position accuracy in a significative way. It may even worsen the precision. The results obtained with the RT products follow the same pattern. The positioning error standard deviation lies around 2 m because the analysis has been performed over the whole trajectory. However, in a dense urban environment, the accuracy of the solution is very poor, mainly because of the bad geometry of the satellites and the multipath effects in the narrow streets (red part of the trajectory depicted in Figure 14b). Keeping only the "sub-urban" part of the trajectory (in blue in Figure 14b), the mean position accuracy is 0.38 m with code and phase measurements and 0.34 m with code, phase, and Doppler measurements in full AR mode. In the Snapshot mode, the code/carrier phase solution leads to an accuracy of 0.48 m and the full code/carrier phase/Doppler solution enables a positioning accuracy of 0.41 m. In both the AR and Snapshot modes, adding the Doppler measurement increases the horizontal accuracy by around 10%. This means that using the Doppler measurement has a greater impact when the environment is harsh. This result follows the theoretical analysis performed by [32]. Regarding the velocity errors, the same conclusions can be drawn. Taking into account the high-rate Doppler measurement leads to an augmentation of the velocity accuracy, especially in the

PPP-Snapshot mode, for which a horizontal standard deviation of 13 cm/s can be obtained with 10 Hz code, phase, and Doppler measurements.

**Table 9.** Horizontal position and velocity accuracies for the kinematic trajectory estimation in Toulouse, France, in several cases, at 1 Hz or 10 Hz, with code and carrier-phase measures only (C/P) or code, carrier-phase, and Doppler-shift measures (C/P/D), with the real-time or the post-processed products. The table also displays the relative difference (rel. diff.) for the different configurations. All position errors are given in meters, and all velocity errors are given in meters per second.

| | | **Position Results** | | | | | | | | | | | |
|---|---|---|---|---|---|---|---|---|---|---|---|---|---|
| | | **Post-Processed Products** | | | | | | **Real-Time Products** | | | | | |
| | | **PPP-AR** | | | **PPP-Snapshot** | | | **PPP-AR** | | | **PPP-Snapshot** | | |
| | | mean | std. | 95% | mean | std. | 95% | mean | std. | 95% | mean | std. | 95% |
| | C/P [m] | 1.51 | 2.25 | 5.95 | 1.59 | 2.36 | 6.42 | 1.54 | 2.55 | 6.62 | 1.86 | 2.58 | 7.25 |
| 1 Hz | C/P/D [m] | 1.51 | 2.31 | 6.41 | 1.62 | 2.22 | 5.84 | 1.48 | 2.33 | 6.00 | 1.90 | 2.47 | 6.77 |
| | Rel. diff. [%] | 0.0 | −2.6 | −7.2 | −1.9 | 6.3 | 9.9 | 4.05 | 9.4 | 10.3 | −2.1 | 4.5 | 7.1 |
| | C/P [m] | 1.47 | 2.59 | 6.94 | 1.54 | 2.46 | 6.96 | 1.64 | 3.89 | 7.46 | 1.85 | 3.89 | 7.81 |
| 10 Hz | C/P/D [m] | 1.14 | 1.66 | 5.17 | 1.20 | 1.72 | 5.26 | 1.14 | 1.76 | 4.80 | 1.27 | 1.90 | 4.87 |
| | Rel. diff. [%] | 29.0 | 56.0 | 34.2 | 28.3 | 43.0 | 32.3 | 43.9 | 121.0 | 55.4 | 45.7 | 104.8 | 60.4 |
| | | **Velocity Results** | | | | | | | | | | | |
| | | **Post-Processed Products** | | | | | | **Real-Time Products** | | | | | |
| | | **PPP-AR** | | | **PPP-Snapshot** | | | **PPP-AR** | | | **PPP-Snapshot** | | |
| | | mean | std. | 95% | mean | std. | 95% | mean | std. | 95% | mean | std. | 95% |
| | C/P [m/s] | 0.43 | 0.49 | 1.30 | 0.85 | 0.97 | 2.87 | 0.54 | 0.62 | 1.79 | 0.93 | 1.06 | 3.08 |
| 1 Hz | C/P/D [m/s] | 0.33 | 0.46 | 1.20 | 0.69 | 0.97 | 2.80 | 0.42 | 0.60 | 1.61 | 0.77 | 1.07 | 3.01 |
| | Rel. diff. [%] | 30.30 | 6.52 | 8.33 | 23.19 | 0.0 | 2.50 | 28.57 | 3.33 | 11.18 | 20.78 | −0.93 | 2.33 |
| | C/P [m/s] | 1.64 | 1.49 | 4.53 | 0.46 | 0.48 | 1.42 | 1.43 | 1.38 | 4.35 | 0.50 | 0.54 | 1.53 |
| 10 Hz | C/P/D [m/s] | 0.89 | 1.27 | 3.66 | 0.07 | 0.12 | 0.24 | 0.57 | 0.91 | 2.58 | 0.08 | 0.13 | 0.27 |
| | Rel. diff. [%] | 84.27 | 17.32 | 23.77 | 557.14 | 300.00 | 491.67 | 150.88 | 51.65 | 68.60 | 525.00 | 315.38 | 466.67 |

Table 10 displays the ambiguity fixing rates in the several cases, according to the use of Doppler measurements or not and the measurement rate. Not surprisingly, processing the Doppler measurements does not improve the ambiguity fixing rates in the PPP-Snapshot mode, since the ambiguities are reset at every epoch and thus do not benefit from the Kalman filter convergence effect. However, when it comes to the PPP-AR mode, the extra-widelane and the widelane fixing rates are increased when using the Doppler measurements, but the narrowlane becomes harder to fix with both the POST and RT products.

**Table 10.** Extra-widelane (EWL), widelane (WL), and narrowlane (NL) ambiguities fixing rates (in %) for the kinematic trajectory estimation in Toulouse, France, in several cases, at 1 Hz or 10 Hz, with code and carrier-phase measures only (C/P) or code, carrier-phase, and Doppler shift measures (C/P/D), with the real-time or the post-processed products.

| | | **Post-Processed Products** | | | | | | **Real-Time Products** | | | | | |
|---|---|---|---|---|---|---|---|---|---|---|---|---|---|
| | | **PPP-AR** | | | **PPP-Snapshot** | | | **PPP-AR** | | | **PPP-Snapshot** | | |
| | | EWL | WL | NL | EWL | WL | NL | EWL | WL | NL | EWL | WL | NL |
| 1 Hz | C/P [%] | 40.28 | 35.12 | 18.04 | 47.89 | 16.71 | 16.95 | 38.08 | 33.76 | 16.39 | 36.96 | 16.65 | 16.90 |
| | C/P/D [%] | 41.58 | 36.05 | 18.53 | 47.89 | 16.71 | 16.95 | 38.09 | 33.98 | 16.39 | 37.02 | 16.65 | 16.90 |
| 10 Hz | C/P [%] | 48.42 | 44.93 | 35.78 | 49.38 | 17.22 | 17.47 | 40.36 | 38.37 | 24.85 | 38.71 | 17.29 | 17.53 |
| | C/P/D [%] | 49.10 | 45.88 | 30.96 | 49.37 | 17.22 | 17.47 | 40.44 | 39.01 | 22.86 | 38.71 | 17.29 | 17.53 |

### 4.2. Academic Comparison

To the authors' knowledge, four independent academic partial validations of the phase biases generated by the CNES PPP-WIZARD demonstrator have been performed.

In the reference [45], the authors made an exhaustive analysis of the CNES phase biases, both at the satellite level and the PPP level. This study showed that the use of all the constellations significantly improved positioning accuracy and convergence time. Unfortunately, the BDS-3 biases were not fully available at that time. The static position root mean square (RMS) values are in accordance with the ones presented in this paper, but the kinematic position RMS values are less precise.

The second one was conducted by NRCan using the following frequency combinations: L1/L2/L5 for GPS, E1/E5a/E6 for Galileo, and L2/L6/L7 for Beidou 2 and 3. Figure 15 shows the PPP-AR positioning errors of (68th percentile) of multiple 5 min runs in various configurations. The improvement brought by Galileo and Beidou phase biases is clearly visible.

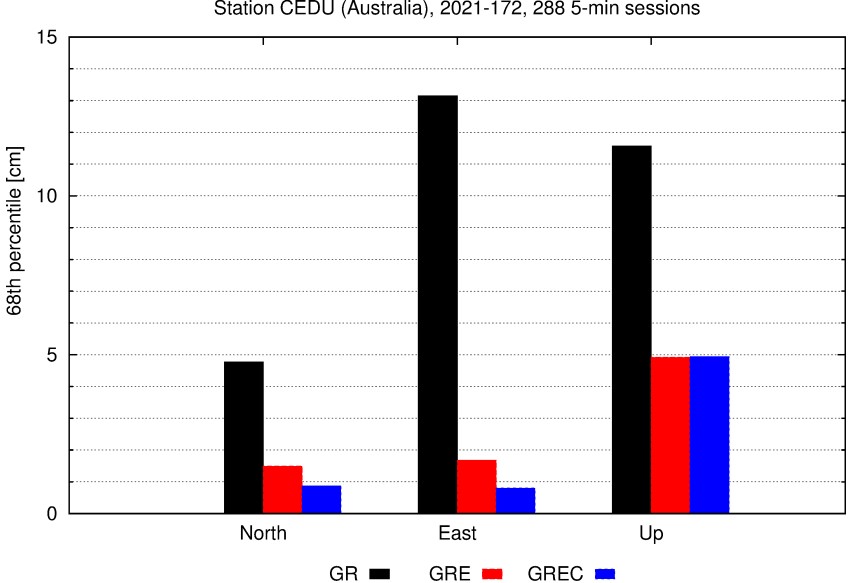

**Figure 15.** NRCan PPP with instantaneous ambiguity resolution. Courtesy: NRCan.

The authors of the reference [46] analyze the benefit of using the real-time phase biases computed by the CNES PPP-WIZARD demonstrator and conclude, as presented in this proposed paper, that using the phase biases improves the positioning accuracy from the PPP-float mode to the PPP-AR mode. In addition to this, they introduced an interpolation method to overcome situations in which phase biases are missing and predict them in such situations.

In the same way, the reference [40] concludes that the CNES phase biases can be used to improve the positioning accuracy in terms of the dispersion of the results around the mean value. On top of that, the mean position errors are smaller, which emphasizes the fact that the errors are more centered around zero.

### 4.3. Commercial Service Comparison

Two comparisons have been performed by companies. The first one is from the Hemisphere company, which uses a proprietary software. Three configurations were tested, summarized in Table 11. A combination of L1/L2/L5, E1/E5a/E6, and B1I/B2I/B3I signals has been applied for GPS, Galileo, and BeiDou, respectively. The following Figure 16a–c shows typical convergence patterns for a selected station (ALBY) and is representative of what can be achieved for each test case:

1. This is the typical convergence in float mode. The process takes about 30 min to attain full convergence.

2. The results are similar to those presented in [30]. With partial ambiguity fixing, 10 cm of accuracy is achieved instantaneously, and fully ambiguity fixing at the centimeter level is obtained after 2 min of convergence.

3. With the addition of Beidou signals, full ambiguity resolution is instantaneous. Note that a better noise combination can be chosen for Beidou (namely B1c/B2a/B3) but has not been used here.

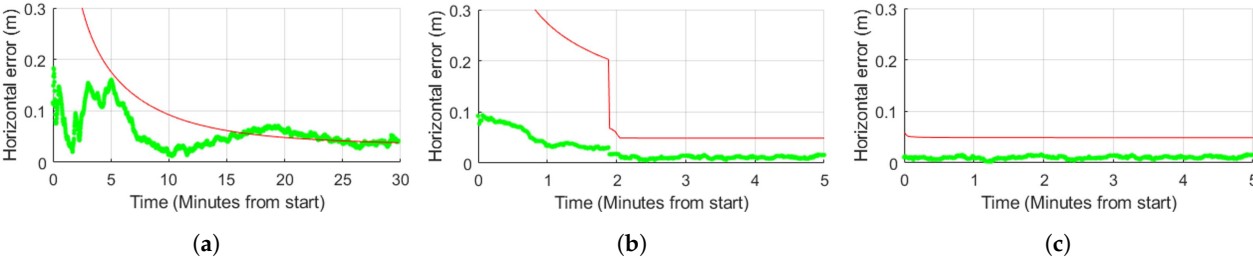

| (**a**) | (**b**) | (**c**) |

**Figure 16.** Positionning results obtained by the Hemisphere company on the ALBY station on 17 June 2021. Courtesy: Hemisphere. (**a**) Hemisphere PPP (float mode). (**b**) Hemisphere PPP (partial AR). (**c**) Hemisphere PPP (full AR).

**Table 11.** Hemisphere PPP-AR processing strategy.

| Test Case | 1 | 2 | 3 |
|---|---|---|---|
| GNSS Measurements | GPS + GAL + BDS | GPS + GAL + BDS | GPS + GAL + BDS |
| Biases applied | Only code biases were applied for the three systems | GPS: code and phase biases, GAL: code and phase biases, BDS: code biases only | Code and phase biases were applied for the three systems |
| Hor. mean [cm] | 5.8 | 1.3 | 1.0 |
| Hor. RMS [cm] | 6.7 | 1.8 | 1.1 |
| Hor 95% [cm] | 13.6 | 3.2 | 1.5 |

Table 11 presents as well the performance results obtained with the three proposed test cases.

The second commercial comparison was conducted by U-Blox. They performed several single epoch tests with a selected set of stations in a three constellations and three bands configuration, and they computed the percentage of epochs where the resulting position was below 5 cm. The results are presented in Figure 17 and show that this criterion is fulfilled a large majority of the time.

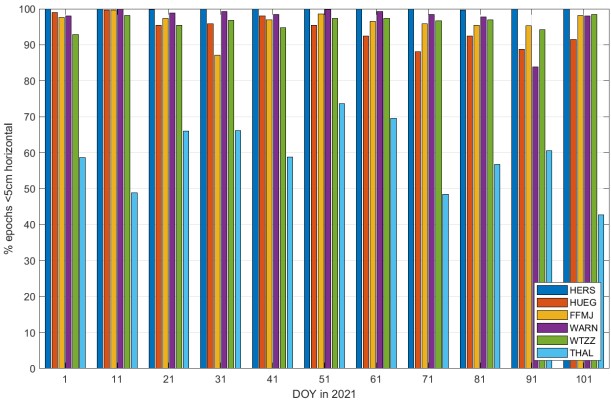

**Figure 17.** Instantaneous convergence below 5 cm. Courtesy: U-Blox.

## 5. Online Positioning Service

CNES has set up a website dedicated to the demonstrator. Its internet address is http://www.ppp-wizard.net (accessed on12 July 2023 ). Figure 18 depicts an image of the main page of the website.

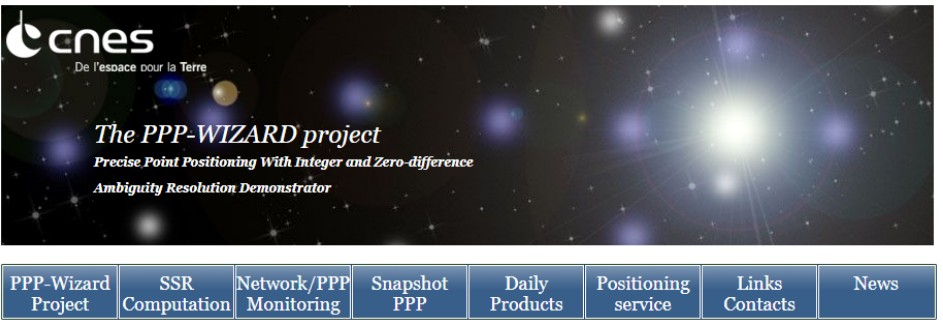

**Figure 18.** The PPP demonstrator webpage.

This website provides all the material to monitor the demonstrator and to perform a positioning with integer ambiguity resolution on the user side:

- a general description of the demonstrator, including descriptions of the new concepts such as the "PPP-Snapshot" mode,
- a set of PPP monitoring stations (for each station, an instance of the PPP-User software is running), the website displays the errors of the obtained solution with respect to an accurate reference and they are updated in real time and reflect the current performance of the demonstrator from the user side,
- links to download daily corrections (real-time or post-processed),
- an online positioning service,
- links to the presentations of the method,
- news of the project.

The main purpose of the online positioning service is to show the quality of the products generated by the demonstrator and also to promote the different ambiguity resolution techniques presented in this paper. This service works like many other online positioning services, such as the NRCAN [17], the JPL [18] and the AUPOS [19] ones. The user uploads a rinex file along with some additional processing choices, and the service computes then returns the precise trajectory. The Figure 19 shows the web interface for the service based on the CNES PPP-WIZARD demonstrator.

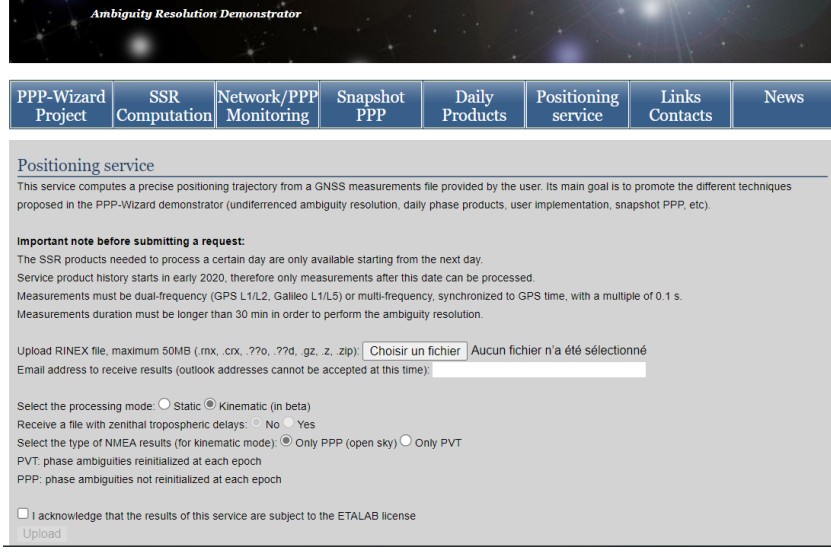

**Figure 19.** Online positioning service web page.

A database of GNSS products is maintained. The products are stored in a daily archive and can be either real-time or post-processed. Real-time products are generated every day at midnight, while post-processed products are generated on a daily basis with a latency of about 7 days. The start date of the archive is 1 January 2020. Each daily product archive contains the following files, which are a self-consistent set of data for further PPP processing: SP3, CLK, BRDM, atx, BIA, OBX. The post-processed or real-time products are chosen when the user sends its request, depending on the age of the measurements in the user rinex file.

The uploaded file is checked for consistency. It must be under 50 MB. It can be of the following formats: rnx, crx, ??d, ??o, gz, or zip. The rinex file must contain at least dual-frequency GPS and/or GLONASS and/or Galileo constellation measurements at a maximal rate of 10 Hz. The maximum time span of the rinex file is 5 days. The header of the file is then checked, in particular the availability of "`MARKER NAME`", "`ANT # / TYPE`", "`ANTENNA: DELTA H/E/N`" fields. Indeed, the exact location of the positioning result depends on these fields.

The rinex file is then sent to the queue of the positioning engine. Several positions can be processed at the same time. If the queue is full, the request is abandoned. The computation uses an instance of the User software [31] in forward/backward mode. Ambiguity resolution is activated. After the computation, a message is sent to the user to summarize the results. If dynamic positioning is requested, a link to download the trajectory (in NMEA format) is available. The Figure 20 gives an example of results for static and dynamic processing.

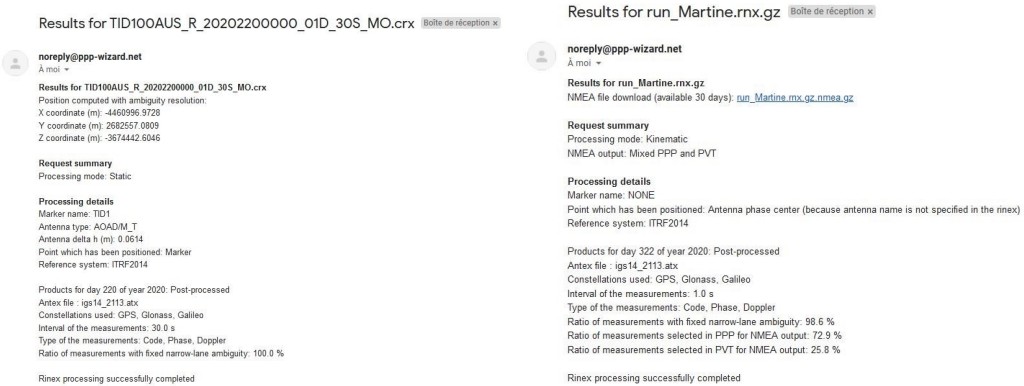

**Figure 20.** Online positioning service result examples.

As the wet tropo is estimated along with the receiver position, it is made available to the users by ticking the appropriate box on the online positioning service webpage.

## 6. Discussion

Even if CNES is a member of the IGS real-time working group, effort has been put recently into the computation of the so-called post-processed code and phase biases using the rapid precise orbit and clock products. It can be seen throughout this study that using these products brings a real benefit from a user's point of view in terms of positioning accuracy. This has been the case in a static scenario with a geodetic station equipped with a good GNSS receiver as well as in a dynamic scenario with a car moving in narrow streets of the city center of Toulouse, France. In addition to this, the use of the POST products shortens the convergence intrinsic to PPP with ambiguity resolution. The improvements brought by these products cannot be obtained for real-time applications. An in-between solution has thus been proposed with the "Snapshot" concept. This positioning technique has been made available thanks to the third and fourth frequencies of Galileo and Beidou. The positioning performances cannot be as good as the PPP-AR mode, but they are still better than the float mode in static conditions and without convergence time. The Snapshot solution could be

improved using meteorological information, such as an external ionosphere provided by an SBAS system.

The fact that independent evaluations of the products computed at CNES can be found in the scientific literature for academic partners or have been made by private companies strengthen the ability for users to improve their positioning performances adopting CNES products or using the provided online positioning service.

## 7. Conclusions

In order to compute the uncombined bias products for GPS L1/L2/L5, Galileo E1/E5a/E5b/E6, Beidou B1/B2a/B2b/B3A, and Glonass G1/G2, CNES has developed a demonstrator called PPP-WIZARD. This demonstrator gathers measurements from 154 ground stations on Earth and disseminates the computed products in SSR and RTCM standards through the IGS. In addition to the network side of the demonstrator, a user side has been set up in order to assess the benefit of using the computing code and phase OSB for positioning in an uncombined and undifferenced manner, leading to the ability to perform PPP with AR. The CNES biases have been compared to existing code and phase OSB products from the School of Geodesy and Geomatics at Wuhan University. The biases have then been used to perform PPP with AR and with the introduced PPP-Snapshot mode in a scenario involving a static receiver in a geodetic station and a moving receiver on a car. The experimental results comparing the real-time products and the post-processed ones showed that the latter drastically improved the positioning accuracy but cannot be used for real-time applications. The academic and commercial assessments conducted by external partners help enforce the quality of the products.

**Author Contributions:** Conceptualization, D.L.; Formal analysis, D.L. and A.B.; Investigation, C.G.; Methodology, D.L.; Project administration, D.L.; Software, E.B., T.J., M.L., N.L. and A.L.; Validation, C.G., A.B., E.B., T.J., M.L., N.L. and A.L.; Writing—original draft, C.G. and D.L.; Writing—review and editing, E.B., T.J., M.L., N.L. and A.L. All authors have read and agreed to the published version of the manuscript.

**Funding:** This research received no external funding.

**Data Availability Statement:** The orbit and clock products as well as the observation data of the ground stations were obtained in the framework of the MGEX experiment from the online archives of the NASA Crustal Dynamics Data Information System (CDDIS): https://cddis.nasa.gov/archive/gnss/data/daily (accessed on 30 June 2023). The CNES real-time and post-processed products can be freely downloaded from http://www.ppp-wizard.net/products/REAL_TIME/ (accessed on 30 June 2023) and http://www.ppp-wizard.net/products/POST_PROCESSED/ (accessed on 30 June 2023). The SGG products have been retrieved from http://igmas.users.sgg.whu.edu.cn/products/download/directory/products/osb/2202 (accessed on 30 June 2023).

**Acknowledgments:** The contributions of colleagues contributing to the IGS services are gratefully acknowledged. We also want to thank NRcan, Hemisphere, and U-Blox for their independent validation of phase biases. A large part of GNSS data used in this paper was obtained from the CNES/IGN REGINA Project (https://regina.cnes.fr/, accessed on 12 July 2023). We also would like to thank the School of Geodey and Geomatics at Wuhan University who gently made their OSB products freely available to the scientific community.

**Conflicts of Interest:** The authors declare no conflict of interest.

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
