# Peer review of "The CNES Solutions for Improving the Positioning Accuracy with Post-Processed Phase Biases, a Snapshot Mode, and High-Frequency Doppler Measurements Embedded in Recent Advances of the PPP-WIZARD Demonstrator"

_remotesensing, doi:10.3390/rs15174231_

Round 1

Reviewer 1 Report

General comments :

- Remove acronyms from the abstract. It would read more intuitively

- State more explicitly the novel aspects in contrast to past publications from CNES. For e.g., how is this work different from the collaboration between CNES and NRCan?

- Additional details needed on the use cases of PPP snapshot mode

Other comments:

Line 1 - "Navigation service" -> Why is navigation capitalized?

Line 2 - "precise point positioning" -> Precise Point Positioning

Line 3 - " Its architecture is composed of three main elements..." -> reword the statement and remove the acronym "OTDS"

Line 34 - " Laurichesse proposed..." Seems awkwardly worked

Line 70 - "difficult environment" was undefined

Line 72 - Unclear what is not optimal in real-time

Line 81 - "Majority of GNSS signals are supported" ... vaguely worded

Line 85 - "very accurate PPP solutions" ... Needs to be quantified. Also, how does it compare to other services?

Line 94 - "indifferenced" -> "undifferenced"

Line 172 - " Doppler measurements for the receiver position estimation" ... Additional detail is needed to explain how the Doppler measurement is being utilized to observe the position states

Figure 2 - black stations were undefined

Section 2.3.3 - The definition of snapshot mode isn't very clear as snapshot mode gives the impression that it's epoch-by-epoch AR fixing but Line 215 speaks about some states have time averaging. Is the ambiguity term re-initialized each epoch? If yes, state this more explicitly. It's stated in line 349 which is too late in the paper. If the ambiguity term is known to be constant over time (assuming no cycle-slips), why would the user be interested in re-initializing the solution each epoch?

 Figure 4 - Label y-axis

Line 258 - Explain to the reader why Glonass was excluded

Figure 7 - Confusing layout, please restructure with clearer labels for (a) and (b)

Figure 8

- Separate code and phase biases

- Why does L2W biased in post-processing?

- Add clearer labels per sub-plot

- Keep column one for "real-time" and column two for "post-processing"

- If the intention is to compare RT and POST, keep y-axis the same

Figure 9 

- why was the Up component excluded?

Line 344 - Is PPP snapshot really an intermediate mode if you're re-intializing each epoch?

Table 6 and 7

 - What is "Vit."?

 - Here the stats for the vertical component is presented but in FIgure 9 it was not presented

Table 9

 - Expected better PPP-AR performance, in contrast to  PPP snapshot

Line 404 - Why rms statistic wasn't included in the tables?

.

Author Response

Please find the reply to the comments in the attached file.

Reviewer 2 Report

The manuscript reports the latest development of PPP-WIZARD demonstrator. The CNES has been a long time a main contributor of PPP real-time application using so-called undifferenced ambiguity resolution approach. This manuscript mainly describes the experimental results. The weak point is that the methodology has not been really clearly presented. In particular, the parameterization of the functional model and the stochastic model are not specified. When I review the manuscript, I have to guess how the authors did the job. Therefore, I strongly suggest that the authors extend the theoretical part and clearly give out, in particular, 1) how the ambiguity integer nature is maintained when undifferenced observations are used; 2) how the parameterization is done for handling the hardware delays in satellite clocks, receiver clocks, ionosphere term and ambiguity terms, as well as in code and phase bias terms in both server side (including when orbit and clock are determined) and receiver side; 3)  how AR solution is computed on the basis of "residual combinations"? In addition, I kept watching the web page of WIZARD project in the past. I seldom observed nice results there. Both the accuracy and reliability are not sufficiently good. There may be reasons of internet or broadcast (assuming that some resets are due to the reason), however, the position time series is not good either when there is no obvious internet issue. Apart from above, there are a few other places that need authors' attentions:

1)    P1, L34: “a integer” -> “an integer”.

2)    P5, “Ioosphere VTEC” -> “Ionosphere VTEC”.

3)    P6, L166 and Eq.(2) are not clear, it is difficult to understand, what is \hat_h^s_r, which is not specified in the context. In addition, where is the satellite offset h^s? Are you sure they are purely corrected by the RTCM/IGS standards’ corrections without containing some hardware biases? Does the ionosphere term contain purely the ionosphere without other hardware biases? Eq.(2) is just a geometry-free model that doe not bring the coordinate parameters in the observation model, however, the manuscript focuses on positioning.

4)    In Fig.3, what do you mean “optimal combination and ambiguity resolution”? "Optimal" must be something inside.

5)    P8, L239-240, the sentence needs really an extensive explanation. The method should be described

6)    P15, suggest to change “Vit.” to “Vel.” .

7)    In Table 6, the configurations for STA and KIN are not clear. What exactly “Position/velocity initial standard deviation” and “Position/velocity model standard deviation” mean? Do they mean “a priori standard deviation” and “spectral density of process noise” for position and velocity parameters? The manuscript should give the functional model and stochastic model in detail so that people can understand what you are talking about in the Table 6. If these standard deviations are set as “zero”, does it mean they are tightly constrained? If the velocities are set for KIN solutions, it means a dynamic model is introduced to the solutions, therefore, we can not claim that they are kinematic solutions any longer. This is at least my understanding, I think the concepts have to be made clearer.  

8)    In Table 7 caption, “TR” -> “RT”

9)  The author sequence is different in PDF file and here in review system. This should be the same.

I suggest that the language needs some moderate improvements and caerful edition. I have included a couple of mistakes in comments to authors.

Author Response

Please find the reply of the comments in the attached file.

Reviewer 3 Report

       The work presents recent advances of the PPP-WIZARD Demonstrator, which is an outstanding open-source GNSS software compatible with real-time CNES products. The code and phase OSB generated from CNES can lead to the ability to perform PPP with AR. An in-between PPP float- and PPP-AR solution in real-time mode has been proposed with the "Snapshot" concept, which is implemented and validated in a static scenario with a geodetic station as well as for a typical urban dynamic environments. The results comparing the real-time products and the post-processed products showed that the latter drastically improves the positioning accuracy but cannot be used for real-time applications. The Snapshot solution can improve positioning accuracy with instantaneous ambiguity resolution, in particular the use of high frequency Doppler measurements and OSB real-time products. I believe this will help enforce the development of the GNSS real-time application. It is recommended that the work is significant and viable, and can be directly accepted by the editor board.

Comment 1:

Will the software PPP-WIZARD release the open-source version of the USER part for GNSS research and education. The open-source software can make the latest advance accessible of PPP- WIZARD to the GNSS community.

Comment 2:

The long-term real-time data from more global GNSS stations should be included and validated for the proposed method and CNES OSB products.

Comment 3:

The results have shown doppler shift measurements have the benefit of improving the positioning performance in urban dynamic environments, especially in high frequency mode. Can you present the statistical result of velocity and ambiguity resolution (AR)?

The English read well.

Author Response

(The authors gave the same response as above.)

Round 2

Reviewer 2 Report

1) In the revised version, should h_s in Eq(2a), (2b) and (4a), (4b) be h^s. The term is also different with the corresponding term in cover letter eq.(3) and (5), there it is written as h_r (left-hand side), while as h_s in (2). Too many typos make the manuscript/answers so difficult to understand.

2) The geometry-based model means that the geometric term \rho_r^s should be written in terms of unit-vector of Line Of Sight multiplying with position vector, because you aim at positioning.

3) L167-L169, the authors should first describe how to solve the rank deficiency issue in network side, rather than the user side. In addition, Image that each site per constellation with four frequency has one receiver clock + three receiver code/phase biases and each satellite has one satellite clock + three satellite code/phase biases, together with ionosphere and ambiguities terms, there are so many parameters, the computation consumption is a big issue that the authors should also mention here.

4) Line258 and Line280,  [?] refers to which reference?

5) L309, In "the computation of the residuals of all...", "residuals" should be observed minus computed (OmC) values, not the residuals. The residuals are derived later after all parameters are estimated and then subtracted from OmC values.

6) Eq. (10) is rank deficiency, the receiver phase bias can not be separated from ambiguity term. How undifferenced ambiguity resolution can be achieved without eliminating the receiver phase bias between satellites?

7) L335, "... is built over each pass of a satellite ...". if so, the solution is running in real-time? if not, does it make sense?

8) The functional models in Eq (1) - (4) are different with the approach used by the IGS products (particularly if you use their clock), however, the authors later on use the GFZ products. If so, the parameterization in Eq(1-4) will not hold any more. In this sense, what you described in theoretic part has no connection with your experiment part. Please clarify.

9) The description in this manuscript is still messy. If the authors want to present clearly what they have been done, it really needs more careful and comprehensive modification. The functional model changes from raw observation approach to Melbourne-Wuebbena to ionosphere-free combination without systematic connection.  

The English needs comprehensive improvements.

Author Response

Please find the answer to your comments in the attached file.
